# MMTOM-QA: Multimodal Theory of Mind Question Answering

## Abstract

Theory of Mind (ToM), the ability to understand people's minds, is an essential ingredient for developing machines with human-level social intelligence. Recent machine learning models, particularly large language models, seem to show some aspects of ToM understanding. However, existing ToM benchmarks use unimodal datasets – either video or text. Human ToM, on the other hand, is more than video or text understanding. People can flexibly reason about another person's mind based on conceptual representations (e.g., goals, beliefs, plans) extracted from *any* available data, which can include visual cues, linguistic narratives, or both. To address this, we introduce a multimodal Theory of Mind question answering (MMToM-QA) benchmark. MMToM-QA comprehensively evaluates machine ToM both on multimodal data and on different kinds of unimodal data about a person's activity in a household environment. To engineer multimodal ToM capacity, we propose a novel method, BIP-ALM (Bayesian Inverse Planning Accelerated by Language Models). BIP-ALM extracts unified representations from multimodal data and utilizes language models for scalable Bayesian inverse planning. We conducted a systematic comparison of human performance, BIP-ALM, and state-of-the-art models, including GPT-4. The experiments demonstrate that large language models and large multimodal models still lack robust ToM capacity. BIP-ALM, on the other hand, shows promising results, by leveraging the power of both model-based mental inference and language models.

## 1 Introduction

Theory of Mind (ToM) is the cognitive ability to ascribe hidden mental states (e.g. goals, beliefs, and desires) to other individuals based on their observed behavior. A hallmark of human social intelligence, ToM serves as the foundation for a wide range of social interactions and a pillar of commonsense reasoning (Lake et al., 2017). Systems designed to safely and productively interact with humans in an open-ended manner, such as assistive robots (e.g., Dautenhahn, 2007; Hadfield-Menell et al., 2016; Puig et al., 2023), AI teachers (e.g., Wang et al., 2021), autonomous vehicles (e.g., Chandra et al., 2020), and cooperative embodied agents (e.g., Bara et al., 2021; Sclar et al., 2022a), would greatly benefit from incorporating ToM reasoning capabilities, either through learning or based on built-in features. The recent advancements in machine learning, especially in the realm of Large Language Models (LLMs), have spurred increased interest in assessing these models' aptitude for ToM reasoning (e.g., Rabinowitz et al., 2018; Kosinski, 2023; Sap et al., 2019; 2022; Ullman, 2023; Shapira et al., 2023; Shu et al., 2021; Moghaddam & Honey, 2023; Nematzadeh et al., 2018; Gandhi et al., 2021). Many of these assessments use either text-based or video-based benchmarks inspired by classic ToM experiments in the cognitive science literature (Wimmer & Perner, 1983).

While the recent benchmarks for testing ToM reasoning in ML models provide well-designed, cognitively informed tools, they share several notable limitations. One such limitation is the dependence on massive training data, which raises the concern that these models work by finding data patterns in a way that deviates from human-like ToM reasoning. This concern is under active discussion (e.g., Ullman, 2023; Sap et al., 2022; Shapira et al., 2023). In this paper, we focus on a different but related limitation: These benchmarks rely on unimodal data, either in the form of videos and animations (e.g., Gandhi et al., 2021), or textual descriptions of actions and environments (e.g., Kosinski,

**VIDEO INPUT**

**TEXT INPUT**

**What's inside the apartment:** … The kitchen is equipped with a microwave, eight cabinets, … Inside the microwave, there is a cupcake. There is a wine glass and an apple on one of the kitchen tables. There are water glasses, a bottle wine, a condiment bottle, and a bag of chips inside the cabinets. …
**Actions taken by Emily:** Emily is initially in the bathroom. She then walks to the kitchen, goes to the sixth cabinet, opens it, subsequently closes it, and then goes towards the fourth cabinet.

**QUESTION**

Which one of the following statements is more likely to be true?

(a) Emily has been trying to get a cupcake. ✔    (b) Emily has been trying to get a wine glass. ✘

Figure 1: **Sketch of the MMToM-QA benchmark**. Each question is associated with a video stream (representative frames highlighting key moments are shown above for illustration) and text input (illustrative text above is shortened for brevity). In the example video, Emily can see the wine glass on one of the kitchen tables (1st frame) and passes by it without picking it up (2nd frame). At the end of the clip (3rd frame), it appears that she could be walking towards the cabinets on the left side of the room; or she might want to check if a goal object is inside the microwave. The text indicates that there are no cupcakes in the cabinets, but there is a cupcake inside the microwave. To confidently choose the correct answer, a model must fuse relevant information from both the video and the text.

2023; Sap et al., 2022). But Theory of Mind reasoning goes beyond merely text comprehension or video understanding. It is about forming a causal model of another person's mind, which connects mental variables to possible actions (Baker et al., 2009; Saxe, 2012; Jara-Ettinger et al., 2016; Jara-Ettinger, 2019). Such a model can infer mental states from either words or vision separately, or fuse the separate information to form a single coherent mental scene. By examining multimodal ToM reasoning, we can both gain insight into the computational models that underlie human ToM and offer a stronger test for current ML models, particularly LLMs.

To systematically evaluate the ability of ML models to infer mental states from multimodal data, we developed a novel Multimodal Theory of Mind Question Answering benchmark (MMToM-QA). As shown in Figure 1, the benchmark includes as input both videos and text describing the activity of a person in a household environment. The benchmark also includes questions associated with different points in each of the videos. The questions refer to the mental states (goals and beliefs) of the person described by the video or text. Each question has two possible options, neither surely true nor surely false, but with one option always significantly more likely to be true given the observations so far. Some questions can be adequately answered based on a single modality (text or video), but some questions require fusing information from both modalities (e.g. understanding the woman's goal in Figure 1). We validated our benchmark through human experiments, showing that people are adept at answering the questions in the benchmark. A human-like machine ToM model should adequately adjust its inferences and associated uncertainties over a person's mental states given the information from both modalities up to the point at which the question is asked, in line with human judgments.

To build a human-like machine ToM model, we propose a novel multimodal ToM model, *Bayesian Inverse Planning Accelerated by Language Models* (BIP-ALM), which combines multimodal data to produce a coherent inference. As illustrated in Figure 3, BIP-ALM first extracts symbolic representations about the physical scene and the actions of the person from both video and text inputs. Using these symbolic representations, BIP-ALM then extends Bayesian inverse planning (BIP) (Baker et al., 2017), a ToM reasoning method that reverse-engineers human ToM but was originally designed for visual data, to reason about the multimodal data. To further accelerate the inference for ToM in real-world scenarios such as household activities in our benchmark, BIP-ALM finetunes a language model using synthetic human activity data and uses it to evaluate the likelihood of hypotheses about the person's belief and goal. By doing so, BIP-ALM takes advantage of the robustness of Bayesian inverse planning, as well as the scalability and open-endedness of language models.

We compared the performance of BIP-ALM and several state-of-the-art models for text QA or multimodal QA, including GPT-4 and Video-LLaMA. We found that existing models, however impressive

in other QA benchmarks, make large and systematic errors in our benchmark, and fail to match human performance. In contrast, BIP-ALM significantly outperforms these models, even when using a very small language model. These findings reveal the limits of current state-of-the-art models and suggest an alternative approach to engineering human-level ToM reasoning.

In summary, our main contributions include (1) the first benchmark for multimodal ToM, (2) a novel ToM reasoning method, BIP-ALM, that combines Bayesian inverse planning and language models to conduct robust yet efficient ToM inference based on multimodal data, and (3) a systematic comparison of different ML models and human ToM.

## 2  RELATED WORK

**Theory of Mind Benchmarks.** Existing ToM benchmarks are based on either videos or text. Visual-based benchmarks such as BIB (Gandhi et al., 2021), AGENT (Shu et al., 2021), or PHASE (Netanyahu et al., 2021) typically use animations of goal-directed agents to evaluate different concepts in ToM. ToMi (Le et al., 2019) and its variants, Adv0-CSFB (Shapira et al., 2023), adapt a classic false belief test, the Sally-Anne test (Wimmer & Perner, 1983), to build text QA benchmarks. Similarly, the episode reasoning benchmark introduced by (Hewitt & Cohen, 2021) has text-based QA tests that ask to select the true hypothesis about a person's knowledge and belief based on a given premise, which extends the Sally-Anne test. Triangle COPA (Gordon, 2016) asks questions about the mental states of agents based on text descriptions of abstract shapes acting like social agents. Moreover, there are QA benchmarks such as Social-IQ (Zadeh et al., 2019) and SocialIQa (Sap et al., 2019) that do not specifically test ToM, but evaluate social intelligence in general. Lastly, there have also been multi-agent challenges (e.g., Sclar et al., 2022b; Puig et al., 2020; 2023) evaluating ToM as part of the tasks. Unlike existing benchmarks, our MMToM-QA evaluates machine ToM on multimodal data to test both goal and belief inference. We also go beyond simple visual and textual stimuli and evaluate ToM using long everyday activities in complex environments. Critically, ToM QAs ask questions about people's mental states (goals and beliefs) based on observable behaviors. This is fundamentally different from VQAs (e.g., Antol et al., 2015; Zellers et al., 2019) which do not require mental state inference. Table 2 in Appendix A provides a detailed comparison.

**Multimodal Question Answering.** There have been several multimodal QA benchmarks developed in recent years (e.g., Talmor et al., 2021; Singh et al., 2021; Sanders et al., 2023; Fu et al., 2023; Li et al., 2023). These benchmarks focus on the ability of models to detect and retrieve relevant information from multimodal inputs (e.g., images, videos, text, tables) to answer factual questions. However, there have not been multimodal QA benchmarks focusing on ToM, an ability fundamentally different from the kind of multimodal information retrieval tested in the existing benchmarks.

**Machine Theory of Mind.** There have been two main approaches to engineering machine ToM: end-to-end methods such as Theory of Mind neural networks (e.g., Rabinowitz et al., 2018), and model-based methods such as Bayesian inverse planning (e.g., Baker et al., 2017; Shu et al., 2021). Both types of approaches focus mostly on unimodal data and simple domains. Recent studies have suggested that machine ToM may also emerge in LLMs such as GPT-4 (Kosinski, 2023; Bubeck et al., 2023). However, more systematic evaluations have shown that apparent ToM capacities in LLMs are not yet as robust as humans (Sap et al., 2022; Shapira et al., 2023; Sclar et al., 2023), and often fail to pass trivial variants of common tests (Ullman, 2023). Our BIP-ALM model builds on the strengths of these different methods. It extends Bayesian inverse planning to fuse multimodal data and conduct inference in complex scenarios with the use of finetuned language models.

## 3  MMTOM-QA BENCHMARK

### 3.1  OVERVIEW

Our benchmark consists of 134 videos of a person looking for daily objects in household environments, in line with cognitive studies examining mental attributions to agents navigating an environment (e.g. Baker et al., 2017). On average, each video has 1,462 frames, depicting 36 human actions. Based on these videos, we constructed 600 questions about a person's goals and beliefs. Each question is paired with a clip of the full activity in a video (as RGB-D frames), as well as a text description of the scene and the actions taken by the person in that clip. All questions have two

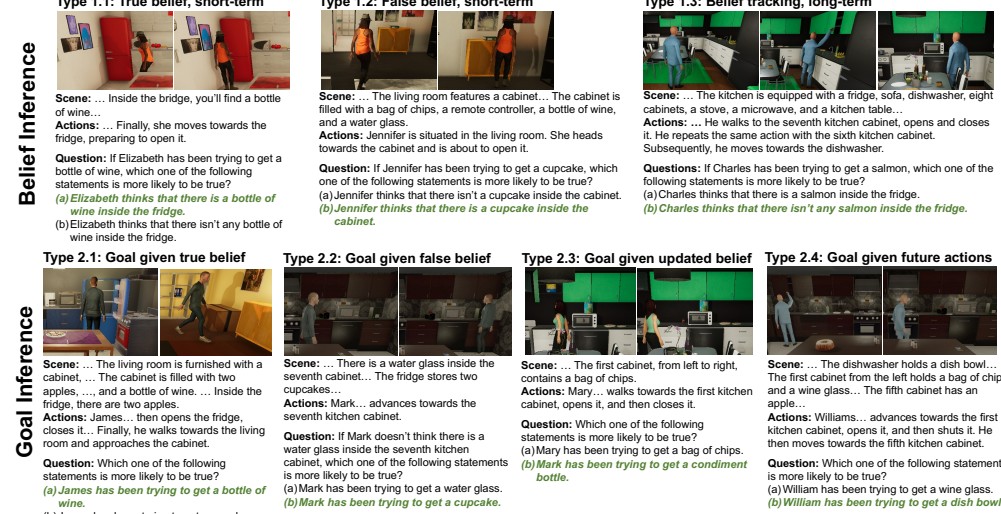

Figure 2: **Question types in MMToM-QA, with examples**. Questions fall into two broad categories, Belief and Goal, with several different question types in each category that span a range of mental reasoning. Each example shows only a few frames and snippets. The options in the green, italic font are correct answers. Note that we simplify the text in the examples for brevity. We provide the full text and the link to the video clip of each example in Appendix E.

choices. The questions are categorized into seven types (see Figure 2), evaluating belief inference and goal inference in rich and diverse situations. Each belief inference type has 100 questions, totaling 300 belief questions; each goal inference type has 75 questions, totaling 300 goal questions. We provide another set of synthetic human behavior data in household environments for model training. This training set includes 1000 procedurally synthesized videos with ground-truth annotations of the scene, objects, goals, and beliefs.

## 3.2 QUESTION TYPES

Questions fall into two categories – belief inference and goal inference. There are several types within each category (Figure 2), evaluating different aspects of multimodal ToM reasoning. Unlike existing ToM evaluation that isolates goal and belief inference, questions in our benchmark require a joint inference of goal and belief, asking about one conditioned on another.

**Type 1.1: True belief, short-term**. In the scenarios this type refers to, a person is about to open a container. The question focuses on an object the person has not seen so far and treats it as a hypothetical goal object. The question is then whether the person believes the object is in the container. This type examines *true belief*; that is, the inference that the person believes the goal is in the container is more consistent with the current action, and it is also the same as the actual world state.

**Type 1.2: False belief, short-term**. This type is similar to Type 1.1, but differs in a significant way: the hypothetical goal object is not inside the container that the person is about to open, so the person has a *false belief*. In this case, the correct answer should still be that it is more likely that the person thinks there is such a goal object inside the container, given the current action, even though they would not find what they want there in reality.

**Type 1.3: Belief tracking, long-term.** In the scenarios this type refers to, the person passes by a container but does not check it. After a while, they have still not found a goal object, but are also not going back to check the container they passed by. This suggests that they do not think the goal object is inside that container. This question tests whether a model can use the long-term observation of past actions to make judgments consistent with history, not just the most recent action.

**Type 2.1: Goal inference given true belief.** This question targets a person's unknown goal. In the scenarios that this type refers to, the person walks towards a container, where there is a hypothetical goal object that the person has not observed so far. The person, on the other hand, *has* observed the other hypothetical goal object but has not picked it up in the past. The correct inference is that it is

more likely that the person wants the goal object that they have not seen so far and thinks it is inside the container (true belief). This type tests whether a model can infer the goal given a true belief.

**Type 2.2: Goal inference given false belief**. This type is similar to type 2.1, but the person has a hypothetical false belief. In these scenarios, a specific object is inside a container that the person is walking towards. However, the question states that this person thinks that there is no such object inside the container (false belief). So, the most likely explanation is that the person's goal is a different object, which they think might be inside the container.

**Type 2.3: Goal inference given updated belief**. Unlike Type 2.1 and 2.2, in a Type 2.3 question, the video does not end with the person walking towards a container. Instead, the person opens the container and then closes it without taking an object from it. The correct inference is that the person's goal is an item not yet seen rather than anything inside the container. To correctly answer this type of question, a model has to infer how a person may update their belief and change their plan (e.g., closing the container without picking anything from it) to pursue the goal accordingly.

**Type 2.4: Goal inference given future actions.** Questions in Type 2.1, 2.2, and 2.3 ask about goals that are consistent with the belief and the *latest* action. In contrast, in Type 2.4, a model needs to consider possible *future* actions as well. Specifically, one of the hypothetical goal objects is an unobserved object at a location that is still far away from the person and is not directly related to the latest action (which is walking to a nearby container). But, in these scenarios, the person is on a path to potentially reach the location of that object. For instance, in the Type 2.4 example illustrated in Figure 2, a person is walking towards the right side of the room, and so they might want to search through all possible locations on the right side, including the dishwasher. This gives rise to the correct answer, dish bowl, which is located inside the dishwasher. As such, Type 2.4 requires a model to reason about the spatial relationships (the locations of objects and the person as well as the person's heading direction) in a visual scene and predict possible future actions for a goal.

## 3.3 PROCEDURAL GENERATION

To generate video and text data, as well as questions of all types at scale for the test set, we designed a procedural generation method. First, we procedurally synthesized a large set of videos in a realistic household embodied simulator, VirtualHome-Social (Puig et al., 2020). As Puig et al. (2020) have demonstrated, such procedural video generation can conveniently create synthetic human activities that are human-like, diverse, and well-annotated (including objects, actions, and ground-truth goals and beliefs). It also alleviates the concerns of cost and privacy that come with collecting real-world human activity videos suitable for ToM reasoning. At each step in a video, we sampled a question type and two opposing hypotheses based on the definition of the type. Finally, we generated the text description and the question based on the ground-truth state, actions, and the sampled hypotheses using GPT-4 to create the text input for the question. Using the same procedural generation, we can synthesize the videos in the training set. We provide more details in Appendix B.5.

## 3.4 EVALUATION PROTOCOL

We can evaluate a model in three conditions: (1) Multimodal QA in which both the video and text inputs are present, (2) Text QA with only the text input, and (3) Video QA with only the video input. We evaluated all models in a zero-shot manner, following prior works on ToM QA evaluation Shapira et al. (2023). Crucially, we do not provide any example QAs during training. We expect a model to learn how a person updates their mental state and acts accordingly in a physical environment from the human behavior data in the training set, and generalize the learned knowledge to answer the questions at test time.

## 4 THE BIP-ALM MODEL

To infer the mental state of a person based on video and text inputs, we propose a novel machine Theory of Mind method, Bayesian Inverse Planning Accelerated by Language Models (BIP-ALM), which builds on Bayesian inverse planning (BIP) (Baker et al., 2017). Prior works have shown that BIP can reverse engineer human ToM reasoning in simple domains. BIP-ALM extends BIP by (1) building unified representations about a scene, a person's actions, and the mental state hypotheses

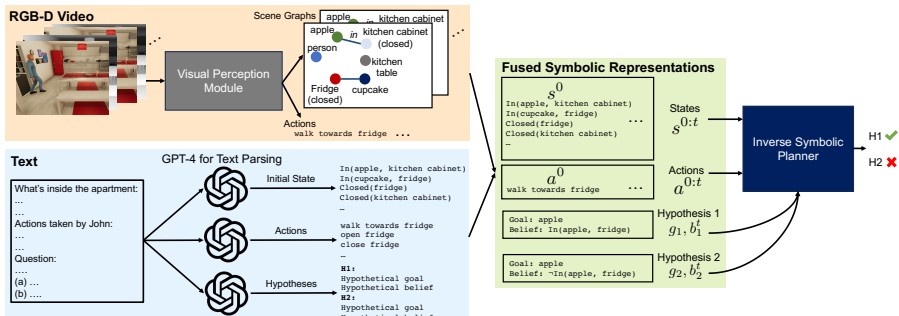

Figure 3: **Overview of our model, BIP-ALM**. For visual, linguistic, and fused information, we show examples of the symbolic representations of states ($s^{1:t}$), actions ($a^{1:t}$), and the two hypotheses about the person's goal ($g_1$ and $g_2$) and belief ($b_1^t$ and $b_2^t$) for a question asked at time step $t$.

from multimodal inputs, and (2) finetuning a language model to efficiently conduct inverse symbolic planning, based on unified symbolic representations.

Figure 3 provides an overview of the method. We first extract symbolic representations of the states and actions from both the video and the text. We then align and fuse representations extracted from different modalities, to form a unified representation of the event and the physical scene. This unified representation allows us to use a principled method to infer a person's mental state based on inputs from *any* given modality. We then use an inverse symbolic planner to compare the two hypotheses extracted from the question and produce the answer. We introduce each module below, and also provide more implementation details in the Appendix C.

### 4.1 UNIFIED SYMBOLIC REPRESENTATIONS

**Visual Perception.** Our visual perception module processes visual data and transforms it into symbolic representations. For each frame, we adopt the method in Blukis et al. (2022) to create a voxel map. We then construct a scene graph based on this voxel map.

**Text Parsing.** We use GPT-4 to parse text, to extract symbolic representations of the initial state as well as subsequent actions. GPT-4 first parses the text into three components – the description of the environment state, the human actions, and the question. Each component is further translated into symbolic representation by different, specifically prompted instances of GPT-4. For state, we generate predicates such as `In(apple, fridge)`. For action, we generate action commands such as `walk towards kitchen`. Finally, we translate the question into two hypotheses about the goal and the belief. For each hypothesis, the goal is represented by the goal object (e.g., `apple`), and the belief is represented by a predicate (e.g., `In(apple, fridge)`), or its negation (`¬In(apple, fridge)`), indicating the location of the goal object in the hypothetical belief.

**Representation Fusion.** The fusion module aligns and integrates information from different input streams. Specifically, we transform the scene graphs from the video input to a set of predicates (similar to those extracted from the text), which describe the spatial relationships between entities and the status of objects. We first form the symbolic representation of the initial state by combining predicates from the video and the text. We then align the actions parsed from the text with the actions detected from the video, and truncate the video frames into several intervals, each corresponding to an action. We term each interval, a time step $t$. Starting from the initial state, we update the state predicates from the previous step with the new predicates obtained from the video frame corresponding to the current step. By doing so, we can construct a symbolic state sequence and a symbolic action sequence, as illustrated in Figure 3. In addition, we form two hypotheses based on the hypothetical goals and beliefs parsed from the question.

### 4.2 INVERSE SYMBOLIC PLANNER

Similar to BIP, we first formulate an agent's behavior as a *forward* generative model using a Partially Observable Markov Decision Process (POMDP) (Kaelbling et al., 1998), defined by the tuple $\langle S, A, T, G, R, \Omega, O, \gamma \rangle$. $s^t \in S$ and $a^t \in A$ are the state and the action at time $t$. $T(s'|s, a)$ are the state transition probabilities. $g \in G$ is a goal, which defines the reward of the agent

$r^t = R(s^t, a^t, g)$. $o^t \in \Omega$ is the agent's observation at $t$ derived following the observation function, $o^t = O(s^t)$. Finally, $\gamma \in (0, 1]$ is a discount factor. The belief of an agent is modeled as a probability distribution over the state $b(s)$. In this work, we factorize the belief of the full state into beliefs about individual objects. For each object, the belief indicates the probability of finding that object at a possible location. Conditioned on both the goal and the belief, a rational agent will take actions based on the optimal policy $\pi(a^t|g, b^t)$ to maximize its return $\sum_{t=0}^{\infty} \gamma^t r^t$.

Given this forward generative model, we can then conduct *inverse* inference about the agent's goal and belief. By assuming a deterministic state transition, we can jointly infer the goal and belief of an agent given observed states and actions as follows:

$$P(g, b^t|s^{1:t}, a^{1:t-1}) \propto P(b^t|b^{t-1}, s^t) \prod_{\tau=0}^{t-1} \pi(a^\tau|g, b^\tau) P(b^\tau|b^{\tau-1}, s^\tau) P(b^0) P(g). \tag{1}$$

Given two hypotheses about goal and belief, $H_1 = \langle g_1, b_1^t \rangle$ and $H_2 = \langle g_2, b_2^t \rangle$, we can evaluate which one is more likely to be true as

$$\frac{P(g_1, b_1^t|s^{1:t}, a^{1:t})}{P(g_2, b_2^t|s^{1:t}, a^{1:t})} = \frac{\pi(a^t|g_1, b_1^t) P(b_1^t|\hat{b}^{t-1}, s^t) \prod_{\tau=0}^{t-1} \pi(a^\tau|g_1, \hat{b}^\tau)}{\pi(a^t|g_2, b_2^t) P(b_2^t|\hat{b}^{t-1}, s^t) \prod_{\tau=0}^{t-1} \pi(a^\tau|g_2, \hat{b}^\tau)}, \tag{2}$$

where $\hat{b}^\tau$ is the estimated belief at a past step $\tau < t$. Since the hypothetical belief in the question is about the belief at the current step, we estimate the belief in the past steps to form a full belief hypothesis. In this work, we assume a uniform distribution for the initial belief. We represent the agent's observation as the subset of the state predicates that the agent can observe, and update the agent's belief accordingly. This gives us an estimated agent belief $\hat{b}^\tau = \hat{b}^\tau(s^\tau)$ at each past step.

Based on Eqn. (2), we need to evaluate (1) the likelihood of the last action $a^t$ given the hypothetical belief and goal ($\pi(a^t|g, b^t)$), (2) the probability of a hypothetical belief at the last step ($P(b^t|\hat{b}^{t-1}, s^t)$), and (3) the likelihood of all past actions given the hypothetical goal and the estimated belief prior to the current step ($\prod_{\tau=0}^{t-1} \pi(a^\tau|g, \hat{b}^\tau)$). The computational bottleneck here is the policy. Conventional methods rely on planning or reinforcement learning to acquire such a policy. Inspired by the recent use of LLMs for decision-making (Huang et al., 2022; Li et al., 2022), we adopt a language model to amortize the policy. In particular, we symbolically represent the belief at each step as a list of possible locations of the corresponding goal object. We then prompt a language model with the symbolic representations of the state $s^t$, goal $g$, and estimated belief $\hat{b}^t$, and generate the likelihood of the observed action $a^t$ based on the output logits. We can finetune the language model on the ground-truth state, belief, goal, and action sequences in the training dataset.

## 5 EXPERIMENTS

### 5.1 BASELINES

**Human baseline.** We conducted a behavioral experiment to validate our questions, and to evaluate human performance. Participants (N=180) were recruited online via Prolific. We randomly sampled 120 questions (20% of all questions) from all types and tested the participants in three conditions: multimodal, text-only, and video-only. Each question in each condition was answered by 5 participants. The experiment was approved by an institutional review board.

**Large language models (LLMs).** We evaluate LLMs on the text-only version of MMToM-QA, including **GPT-4** (OpenAI, 2023), **GPT-3.5**, **GPT-J** (6B) (Wang & Komatsuzaki, 2021), and **LLaMA 2** (7B) (Touvron et al., 2023). For *all* LLMs, we prompt them with the text input.

**Large multimodal models (LMMs).** We evaluate state-of-the-art LLMs – **InstructBLIP** (Dai et al., 2023) and **Video-LLaMA 2** (Zhang et al., 2023) – on both multimodal and video-only versions of MMToM-QA. For *all* LMMs, we uniformly sample a few frames from each video following prior works (Dai et al., 2023). In particular, we sample 16 frames and 30 frames from each video for InstructBLIP and Video-LLaMA 2 respectively. We use the largest versions of both models, Vicuña-13B for InstructBLIP and LLaMA-2-13B-Chat for Video-LLaMA 2. To test GPT-4 in both multimodal and video-only conditions, we create a baseline, **GTP-4 with captions**, by appending video captions generated by Video-LLaMA 2 to the prompt for GPT-4.

In the experiments, we finetuned GPT-J (6B) and LLaMA 2 (7B) and created two BIP-ALM models: **BIP-ALM w/ GPT-J** and **BIP-ALM w/ LLaMA 2**.

Table 1: Human and model performance in each question type.

| | Method | Belief Inference | | | | Goal Inference | | | | | All |
|---|---|---|---|---|---|---|---|---|---|---|---|
| | | 1.1 | 1.2 | 1.3 | All | 2.1 | 2.2 | 2.3 | 2.4 | All | |
| **Multimodal** | Human | 95.8 | 96.7 | 100 | 97.5 | 90.0 | 91.7 | 83.3 | 88.9 | 88.5 | 93.0 |
| | InstructBLIP | 62.0 | 52.0 | 32.0 | 48.7 | 46.7 | 29.3 | 42.7 | 60.0 | 44.7 | 46.7 |
| | Video-LLaMA 2 | 36.0 | 38.0 | 52.0 | 42.0 | 36.0 | 41.3 | 30.7 | 45.3 | 38.3 | 40.2 |
| | GPT-4 with captions | **98.0** | 37.0 | 84.0 | 73.0 | 49.3 | 58.7 | 1.3 | 48.0 | 39.3 | 56.2 |
| | BIP-ALM w/ GPT-J | 90.0 | **69.0** | **86.0** | **81.7** | **68.0** | **78.7** | 56.0 | 73.3 | 69.0 | 75.3 |
| | BIP-ALM w/ LLaMA 2 | 88.0 | 68.0 | 85.0 | 80.3 | 62.7 | 77.3 | **72.0** | **80.0** | **73.3** | **76.7** |
| **Text only** | Human | 96.0 | 95.8 | 81.3 | 91.0 | 85.8 | 76.7 | 65.0 | 68.3 | 74.0 | 82.5 |
| | GPT-4 | **97.0** | 12.0 | 77.0 | 62.0 | 48.0 | 42.7 | 2.7 | 42.7 | 34.0 | 48.0 |
| | GPT-3.5 | 81.0 | 11.0 | 39.0 | 43.7 | 46.7 | 16.0 | 21.3 | 48.0 | 33.0 | 38.3 |
| | GPT-J | 56.0 | 53.0 | 38.0 | 49.0 | 52.0 | 50.7 | **50.7** | 56.0 | 52.3 | 59.7 |
| | LLaMA 2 | 64.0 | 55.0 | 50.0 | 56.3 | 49.3 | 48.0 | 41.3 | 38.7 | 44.3 | 50.3 |
| | BIP-ALM w/ GPT-J | 88.0 | **69.0** | 88.0 | 81.7 | **77.3** | 68.0 | 30.7 | **70.7** | **61.7** | **71.7** |
| | BIP-ALM w/ LLaMA 2 | 89.0 | 68.0 | **90.0** | **82.3** | 54.7 | 66.7 | 50.7 | 62.7 | 58.7 | 70.5 |
| **Video only** | Human | 69.1 | 64.3 | 86.4 | 73.3 | 58.5 | 60.0 | 76.7 | 63.3 | 64.6 | 68.9 |
| | InstructBLIP | 56.0 | 50.0 | 42.0 | 49.3 | 56.0 | 45.3 | 54.7 | 53.3 | 52.3 | 50.8 |
| | Video-LLaMA 2 | 24.0 | 32.0 | 67.0 | 41.0 | 50.7 | 45.3 | **56.0** | 52.0 | 51.0 | 46.0 |
| | GPT-4 with captions | 58.0 | 21.0 | 41.0 | 40.0 | 45.3 | 38.3 | 42.6 | 37.3 | 40.9 | 40.5 |
| | BIP-ALM w/ GPT-J | 63.0 | 57.0 | **72.0** | **64.0** | 45.3 | **62.7** | 50.7 | **62.7** | 55.3 | 59.7 |
| | BIP-ALM w/ LLaMA 2 | **69.0** | **63.0** | 60.0 | **64.0** | **62.7** | 54.7 | 53.3 | **62.7** | **58.3** | **61.2** |

## 5.2 RESULTS

We summarize the results in Table 1. On the multimodal version, humans achieve 93% accuracy averaging across question types. For each tested question, the majority of the participants chose the correct answer, validating our question designs. When only given unimodal data, human performance overall drops in accuracy, with video-only questions being harder to answer than text-only. While most questions suffer, the performance of text-based reasoning on the false-belief/true-belief remains the same. This is of note as these questions have been the main target of text ToM QAs.

The baselines show performance that is close to random guessing across all types, except GPT-4. GPT-4 reaches human-level accuracy on Type 1.1 and shows competitive performance on Type 1.3, in both text QAs and multimodal QAs (combined with the captions from Video-LLaMA 2). However, it also makes systematic mistakes in harder questions that involve false beliefs (Type 1.2). This suggests that GPT-4 can understand the true world state from the text but confuses belief with the true world state. Hence it fails to recognize false beliefs. GPT-4 also struggles with goal inference. Its accuracy on Type 2.3 is particularly low. We hypothesize that this is because GPT-4 mistakenly thinks that the goal has to be one of the objects inside the container the person opens and fails to recognize that the person updates the belief after checking inside the container.

Our BIP-ALM models significantly outperform all baselines in all three conditions by a large margin. Even without fine-tuning, as the ablated study in Appendix B.1 shows, our model with the pretrained GPT-J and LLaMA 2 can already achieve better results than using pretrained LLMs alone, even outperforming a much larger model (GPT-4). In addition, our models have no exposure to modality-specific training – they are only trained on symbolic representations of the training data. This allows them to flexibly conduct ToM reasoning with any unimodal or multimodal data.

## 5.3 ADDITIONAL EVALUATION

**Effect of model sizes.** We report the performance of open-sourced models (LLaMA 2, InstructBLIP, and Video-LLaMA 2) with different model sizes in Table 5 in Appendix B.1. We found that all baselines performed no better than chance, regardless of the model sizes.

**Effect of few-shot prompting or chain-of-thought.** As shown in Table 4 in Appendix B.1, we found no meaningful improvement for almost all baselines after using different few-shot or chain-of-thought Kojima et al. (2022) prompting. They still perform no better than chance. The only exception is GPT-4 in the text-only condition, which has an improvement in simple types (e.g., Type 1.3) with few-shot prompting but still performs poorly on harder types, e.g., false belief (Type 1.2).

**Finetuning baselines.** We finetuned Video-LLaMA 2 (13B) on our training set for video instruction tasks following Zhang et al. (2023). As Table 6 shows, the finetuned Video-LLaMA 2 performs moderately better in a few simpler question types, most notably true belief (Type 1.1), but its overall performance is still not better than chance, unlike our method.

**Generalization.** We created an additional test set, the human test set, for generalization evaluation. It has 40 videos and 120 questions. To generate the videos in this set, we used 2 new apartments unseen in the training set and the main test set. We recruited 3 participants who had no prior exposure to the system to control the avatar to reach assigned goals via the human interface so that we could collect *real human belief updates* and *human actions*. We then used the same method to generate the questions. We report the model performance on this human test set in Table 7. The results show that our method can generalize to both real human behavior and unseen physical environments.

## 6 DISCUSSION & CONCLUSION

We presented Multimodal Theory of Mind Question Answering (MMToM-QA), the first multimodal benchmark for machine ToM. The results of our human experiment verified the design of the questions and provided a human baseline. Comparison of human performance across different modalities reveals the kinds of modalities necessary for answering each type of question. We conducted a systematic evaluation of state-of-the-art LLMs and LMMs and compared them with our method. We summarize several key findings and discuss the limitations and future work as follows.

**What information necessary for ToM can we get from each modality?** From a video, a model gets the dynamic state change at each step as well as what objects the agent is walking towards and is passing by at a given step. A model needs this information to determine the agent's expected action plans given a hypothesis about the belief and the goal. From the text, a model gets ground truth information about the initial state. Because of the partial observations caused by the limited camera view and occlusion, the text provides additional state information that is sometimes unavailable in the video. A model requires information about the true world state to determine whether an agent has a true belief or false belief as well as what objects the agent has observed so far.

**What do current LLMs and LMMs understand about ToM?** We found strong performance from GPT-4 on questions that only require retrieving information about the true world state. However, the results overall show GPT-4 cannot still reason about the mental state of a person and track the change in the mental state over time. We found more specifically that GPT-4 has poor judgment on goals, which was not discussed in prior literature due to the lack of goal inference tests in existing text-based ToM benchmarks. Finally, we found that current LMMs can produce useful captions about people's activities, which can be seen by the marginal performance improvement of GPT-4 with captions compared to only prompting GPT-4 with text input (i.e., GPT-4 in the text-only condition). However, on their own, current LMMs' ToM reasoning capabilities appear to be inferior to GPT-4.

**What are the successes and failures of BIP-ALM?** Instead of directly mapping the multimodal inputs to beliefs and goals, BIP-ALM conducts model-based inference by imaging possible actions given a hypothetical mental state and state context via language models. This not only results in a much better performance in the main test set but also enables the method to generalize to real human behavior in unseen environments. These successes suggest that machine ToM can benefit from (1) the modality-invariance of symbolic representations, (2) the robustness of inverse planning, and (3) the scalability of language models. We also observed several failures. First, BIP-ALM cannot imagine missing state information from videos. Second, it does not have motion planning, which is crucial for Type 2.4, as the model has to imagine possible future paths. Finally, the action likelihood estimated by the LMs sometimes could be inaccurate due to LMs' occasional failures in planning.

**Limitations and future work**. First, MMToM-QA only includes videos of people looking for objects in household environments. In the future, we would like to extend this to more diverse scenarios. Second, in addition to goals and beliefs, ToM also includes reasoning about desires, emotions, and constraints, which we intend to incorporate in a future version of the benchmark. Finally, we intend to enrich the representations in BIP-ALM with broader relations and predicates, extending its reach even further to more complex scenes and human behaviors. This could potentially achieved by finetuning larger language models in broader datasets. Such datasets can be collected in diverse simulators; they can also be crowdsourced (text descriptions of real-world human behaviors).

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

# A    COMPARISON OF THEORY OF MIND BENCHMARKS

Table 2: A comparison of Theory of Mind benchmarks.

| Dataset | Tested Concepts | Test Size | Modality | Generation | Evaluation |
|---|---|---|---|---|---|
| **ToMi** (Le et al., 2019) | False belief | 400 | Text | Templates | Multiple choice Q&A |
| **epistemic reasoning** (Hewitt & Cohen, 2021) | Knowledge, beliefs | 2000 | Text | Templates | True or false judgement |
| **Adv-CSFB** (Kosinski, 2023) | False belief | 183 | Text | Hand-designed | Multiple choice filling in the blanks |
| **Triangle COPA** (Gordon, 2016) | Social interaction | 100 | Text | Hand-designed | Multiple choice Q&A |
| **BIB** (Gandhi et al., 2021) | Goal preferences, efficient actions, constraints, instrumental actions | 5,000 | Video | Procedural generation | Surprise rating |
| **AGENT** (Shu et al., 2021) | Goal preferences, efficient actions, unobserved constraints, cost-reward trade-off | 960 | Video | Procedural generation | Surprise rating |
| **PHASE** (Netanyahu et al., 2021) | Goals, relationships | 100 | Video | Procedural generation | Multiple choice recognition |
| **MMToM-QA** (Our benchmark) | Beliefs, goals | 600 | Text and video | Procedural generation | Multiple choice Q&A |

We provide a comparison of Theory of Mind benchmarks in Table 2, summarizing the evaluated ToM concepts, the size of the test set, available modalities of the inputs, the generation method, and the evaluation for each benchmark. From the table, we can see that our benchmark is the only one that provides multimodal inputs. Additionally, commonly used text-based ToM QA benchmarks do not evaluate goal inference, whereas questions in our benchmark ask about both goals and beliefs.

# B    BENCHMARK DETAILS

## B.1    MORE QUANTITATIVE RESULTS

We conducted an ablated study to show the effect of finetuning language models for BIP-ALM. As Table 3 shows, finetuning GPT-J and LLaMA 2 significantly boosts the performance of BIP-ALM. It is also interesting to see that even before finetuning, BIP-ALM with pretrained GPT-J or LLaMA can already outperform much larger models including GPT-4. This further demonstrates the advantage of conducting inverse symbolic planning for multimodal ToM reasoning.

We have also evaluated the effect of few-shot or chain-of-thought prompting (Table 4), the effect of model sizes (Table 5), the effect of finetuning baseline models on our training set (Table 6), and the generalization of models on the human test set with *real* human behaviors in *unseen* environments (Table 7).

## B.2    QUALITATIVE RESULTS

Figure 4 demonstrates how the inverse symbolic planner enabled by the language model in our BIP-ALM model can estimate the likelihood of a given hypothesis. In Figure 4**A**, given the goal of getting a water glass, our model reasons that Elizabeth is more likely to open the microwave if

Table 3: Results of the ablated study. "w/o FT" indicates ablated models in which our model uses pretrained language models without finetuning. "MM" represents the multimodal condition.

| | Method | Belief Inference | | | | Goal Inference | | | | | All |
|---|---|---|---|---|---|---|---|---|---|---|---|
| | | 1.1 | 1.2 | 1.3 | All | 2.1 | 2.2 | 2.3 | 2.4 | All | |
| **MM** | Ours GPT-J (w/o FT) | 84.0 | 63.0 | 92.0 | 79.7 | 58.7 | 64.0 | 16.0 | 65.3 | 51.0 | 65.3 |
| | Ours GPT-J | 90.0 | 69.0 | 86.0 | 81.7 | 68.0 | 78.7 | 56.0 | 73.3 | 69.0 | 75.3 |
| | Ours LLaMA 2 (w/o FT) | 56.0 | 46.0 | 96.0 | 66.0 | 66.7 | 48.0 | 29.3 | 69.3 | 53.3 | 59.7 |
| | Ours LLaMA 2 | 88.0 | 68.0 | 85.0 | 80.3 | 62.7 | 77.3 | 72.0 | 80.0 | 73.3 | 76.7 |
| **Text** | Ours GPT-J (w/o FT) | 76.0 | 61.0 | 90.0 | 75.7 | 44.0 | 58.7 | 26.7 | 56.0 | 46.3 | 61.0 |
| | Ours GPT-J | 88.0 | 69.0 | 88.0 | 81.7 | 77.3 | 68.0 | 30.7 | 70.7 | 61.7 | 71.7 |
| | Ours LLaMA 2 (w/o FT) | 66.0 | 53.0 | 98.0 | 72.3 | 57.3 | 41.3 | 30.7 | 65.3 | 48.7 | 60.5 |
| | Ours LLaMA 2 | 89.0 | 68.0 | 90.0 | 82.3 | 54.7 | 66.7 | 50.7 | 62.7 | 58.7 | 70.5 |
| **Video** | Ours GPT-J (w/o FT) | 57.0 | 36.0 | 77.0 | 56.7 | 56.0 | 60.0 | 36.0 | 54.7 | 51.7 | 54.2 |
| | Ours GPT-J | 63.0 | 57.0 | 72.0 | 64.0 | 45.3 | 62.7 | 50.7 | 62.7 | 55.3 | 59.7 |
| | Ours LLaMA 2 (w/o FT) | 51.0 | 33.0 | 75.0 | 53.0 | 45.3 | 72.0 | 41.3 | 50.7 | 52.3 | 52.7 |
| | Ours LLaMA 2 | 69.0 | 63.0 | 60.0 | 64.0 | 62.7 | 54.7 | 53.3 | 62.7 | 58.3 | 61.2 |

Figure 4: Examples of how BIP-ALM evaluates the likelihood of different hypotheses via the action likelihood estimation from the language model. The results here are based on BIP-ALM with fine-tuned LLaMA 2. The green option in each example is the correct answer and BIP-ALM selects the correct answers in both cases. The blue panels show the likelihood ratio estimated by the language model at a certain step for each example, explaining how BIP-ALM can come to the correct conclusions by conducting inverse planning via a language model. (**A**) It is more likely for Elizabeth to open the microwave if she believes that there is a water glass inside the microwave and that she wants to get a water glass, even though there is not any water glass inside the microwave (i.e., she has a false belief)). (**B**) The likelihood of hypothesis will change after the model observes more actions.

she believes that there is a water glass inside the microwave, even though there is not any water glass inside the microwave according to the true world state. By imagining reasonable actions conditioned on hypothesized mental states, our model can successfully infer that Elizabeth has a false belief. In contrast, GPT-4 selects (b) as the more likely option, failing to recognize the false belief. Figure 4**B** depicts how the likelihood of a hypothesis changes after the model observes different actions. Specifically, in this case, the model first thinks that it is more likely that Karen is going to open the oven if the goal is to get a plate because there is a plate inside the oven. However, if the goal is truly to get a plate, at the next step, Karen should not close the oven but pick up the plate instead. If the goal is not to get a plate but a salmon, on the other hand, then Karen should close the oven and continue to look for salmon in other places. Therefore, the fact that Karen closes the oven without picking up the plate suggests that it is unlikely that her goal is to get a plate and that it is still quite possible that she wants to get a salmon instead. The action likelihood ratios estimated by

Table 4: Results of baselines using few-shot prompting or chain-of-thought (CoT).

| | Method | Belief Inference | | | | Goal Inference | | | | | All |
|---|---|---|---|---|---|---|---|---|---|---|---|
| | | 1.1 | 1.2 | 1.3 | All | 2.1 | 2.2 | 2.3 | 2.4 | All | |
| **Multimodal** | InstructBLIP | 62.0 | 52.0 | 32.0 | 48.7 | 46.7 | 29.3 | 42.7 | 60.0 | 44.7 | 46.7 |
| | InstructBLIP (w/ 1-shot) | 42.0 | 49.0 | 53.0 | 48.0 | 50.7 | 33.3 | 44.0 | 46.7 | 43.7 | 45.8 |
| | InstructBLIP (w/ 2-shot) | 31.0 | 26.0 | 55.0 | 37.3 | 50.7 | 30.7 | 41.3 | 38.7 | 40.3 | 38.8 |
| | InstructBLIP (w/ CoT) | 68.0 | 66.0 | 39.0 | 57.7 | 45.3 | 26.7 | 45.3 | 42.7 | 40.0 | 48.8 |
| | GPT-4 with captions | 98.0 | 37.0 | 84.0 | 73.0 | 49.3 | 58.7 | 1.3 | 48.0 | 39.3 | 56.2 |
| | GPT-4 with captions (w/ 1-shot) | 94.0 | 22.0 | 55.0 | 57.0 | 49.3 | 85.3 | 5.3 | 50.7 | 47.7 | 52.3 |
| | GPT-4 with captions (w/ 2-shot) | 95.0 | 14.0 | 89.0 | 66.0 | 41.3 | 93.3 | 2.7 | 46.7 | 46.0 | 56.0 |
| | GPT-4 with captions (w/ CoT) | 59.0 | 49.0 | 78.0 | 62.0 | 56.1 | 56.5 | 2.6 | 40.0 | 38.8 | 50.4 |
| | Video-LLaMA 2 | 36.0 | 38.0 | 52.0 | 42.0 | 36.0 | 41.3 | 30.7 | 45.3 | 38.3 | 40.2 |
| | Video-LLaMA 2 (w/ 1-shot) | 45.0 | 45.0 | 37.0 | 42.3 | 46.7 | 48.0 | 42.7 | 48.0 | 46.3 | 44.3 |
| | Video-LLaMA 2 (w/ 2-shot) | 57.0 | 42.0 | 36.0 | 45.0 | 45.3 | 36.0 | 41.3 | 53.3 | 44.0 | 44.5 |
| | Video-LLaMA 2 (w/ CoT) | 30.0 | 13.0 | 43.5 | 28.8 | 25.0 | 24.8 | 15.3 | 12.8 | 19.5 | 24.5 |
| **Text only** | GPT-4 | 97.0 | 12.0 | 77.0 | 62.0 | 48.0 | 42.7 | 2.7 | 42.7 | 34.0 | 48.0 |
| | GPT-4 (w/ 1-shot) | 99.0 | 40.0 | 86.0 | 75.0 | 61.3 | 49.3 | 2.7 | 54.7 | 42.0 | 58.5 |
| | GPT-4 (w/ 2-shot) | 99.0 | 39.0 | 96.0 | 78.0 | 72.0 | 100 | 0 | 45.3 | 54.3 | 66.2 |
| | GPT-4 (w/ CoT) | 97.0 | 13.0 | 82.0 | 64.0 | 49.3 | 5.3 | 2.7 | 44.0 | 25.3 | 44.7 |
| | GPT-3.5 | 81.0 | 11.0 | 39.0 | 43.7 | 46.7 | 16.0 | 21.3 | 48.0 | 33.0 | 38.3 |
| | GPT-3.5 (w/ 1-shot) | 100 | 25.0 | 13.0 | 46.0 | 49.3 | 2.7 | 32.0 | 37.3 | 30.3 | 38.1 |
| | GPT-3.5 (w/ 2-shot) | 100 | 16.0 | 16.0 | 44.0 | 61.3 | 36.0 | 13.3 | 40.0 | 37.7 | 40.8 |
| | GPT-3.5 (w/ CoT) | 82.0 | 11.0 | 40.0 | 44.3 | 45.3 | 10.7 | 21.3 | 46.7 | 31.0 | 37.7 |
| | GPT-J | 56.0 | 53.0 | 38.0 | 49.0 | 52.0 | 50.7 | 50.7 | 56.0 | 52.3 | 59.7 |
| | GPT-J (w/ 1-shot) | 44.0 | 47.0 | 54.0 | 48.3 | 53.3 | 52.0 | 53.3 | 58.7 | 54.3 | 51.3 |
| | GPT-J (w/ 2-shot) | 44.0 | 47.0 | 54.0 | 48.3 | 53.3 | 52.0 | 53.3 | 58.7 | 54.3 | 51.3 |
| | GPT-J (w/ CoT) | 59.0 | 58.0 | 41.0 | 52.7 | 53.3 | 48.0 | 36.0 | 42.7 | 45.0 | 48.8 |
| | LLaMA 2 | 64.0 | 55.0 | 50.0 | 56.3 | 49.3 | 48.0 | 41.3 | 38.7 | 44.3 | 50.3 |
| | LLaMA 2 (w/ 1-shot) | 66.0 | 65.0 | 31.0 | 54.0 | 48.0 | 34.7 | 45.3 | 44.0 | 43.0 | 48.5 |
| | LLaMA 2 (w/ 2-shot) | 48.0 | 51.0 | 50.0 | 49.7 | 46.7 | 48.0 | 46.7 | 50.7 | 48.0 | 48.8 |
| | LLaMA 2 (w/ CoT) | 56.0 | 53.0 | 46.0 | 51.7 | 46.7 | 48.0 | 46.7 | 41.3 | 45.7 | 48.7 |
| **Video only** | InstructBLIP | 56.0 | 50.0 | 42.0 | 49.3 | 56.0 | 45.3 | 54.7 | 53.3 | 52.3 | 50.8 |
| | InstructBLIP (w/ 1-shot) | 63.0 | 60.0 | 32.0 | 51.7 | 56.0 | 41.3 | 52.0 | 62.7 | 53.0 | 52.3 |
| | InstructBLIP (w/ 2-shot) | 67.0 | 52.0 | 21.0 | 46.7 | 54.7 | 56.0 | 58.7 | 58.7 | 57.0 | 51.8 |
| | InstructBLIP (w/ CoT) | 51.0 | 54.0 | 62.0 | 55.7 | 46.7 | 48.0 | 46.7 | 42.7 | 46.0 | 50.8 |
| | GPT-4 + captions | 58.0 | 21.0 | 41.0 | 40.0 | 45.3 | 38.3 | 42.6 | 37.3 | 40.9 | 40.5 |
| | GPT-4 + captions (w/ 1-shot) | 66.0 | 65.0 | 36.0 | 55.7 | 30.7 | 28.0 | 32.0 | 25.3 | 29.0 | 42.3 |
| | GPT-4 + captions (w/ 2-shot) | 79.0 | 38.0 | 24.0 | 47.0 | 38.7 | 74.7 | 60.0 | 33.3 | 51.7 | 49.3 |
| | GPT-4 + captions (w/ CoT) | 52.0 | 27.0 | 47.0 | 42.0 | 50.0 | 53.5 | 38.5 | 41.6 | 45.9 | 44.0 |
| | Video-LLaMA 2 | 24.0 | 32.0 | 67.0 | 41.0 | 50.7 | 45.3 | 56.0 | 52.0 | 51.0 | 46.0 |
| | Video-LLaMA 2 (w/ 1-shot) | 54.0 | 49.0 | 45.0 | 49.3 | 46.7 | 45.0 | 46.0 | 41.3 | 44.8 | 47.0 |
| | Video-LLaMA 2 (w/ 2-shot) | 47.0 | 49.0 | 53.0 | 49.7 | 48.0 | 49.3 | 49.3 | 50.7 | 49.3 | 49.5 |
| | Video-LLaMA 2 (w/ CoT) | 28.6 | 24.4 | 26.7 | 25.6 | 11.1 | 45.6 | 11.7 | 26.8 | 23.8 | 24.7 |

Table 5: Results of baselines with different model sizes.

| | Method | Belief Inference | | | | Goal Inference | | | | | All |
|---|---|---|---|---|---|---|---|---|---|---|---|
| | | 1.1 | 1.2 | 1.3 | All | 2.1 | 2.2 | 2.3 | 2.4 | All | |
| **MM** | InstructBLIP (7B) | 49.0 | 53.0 | 62.0 | 54.7 | 52.0 | 48.0 | 50.7 | 37.7 | 47.0 | 50.8 |
| | InstructBLIP (13B) | 62.0 | 52.0 | 32.0 | 48.7 | 46.7 | 29.3 | 42.7 | 60.0 | 44.7 | 46.7 |
| | Video-LLaMA 2 (7B) | 21.9 | 12.5 | 25.0 | 19.8 | 21.4 | 10.5 | 17.7 | 16.7 | 15.6 | 18.2 |
| | Video-LLaMA 2 (13B) | 36.0 | 38.0 | 52.0 | 42.0 | 36.0 | 41.3 | 30.7 | 45.3 | 38.3 | 40.2 |
| **Text** | LLaMA 2 (7B) | 64.0 | 55.0 | 50.0 | 56.3 | 49.3 | 48.0 | 41.3 | 38.7 | 44.3 | 50.3 |
| | LLaMA 2 (13B) | 44.0 | 47.0 | 54.0 | 48.3 | 48.0 | 37.3 | 49.3 | 60.0 | 48.7 | 48.5 |
| **Video** | InstructBLIP (7B) | 47.0 | 41.0 | 59.0 | 49.0 | 45.3 | 38.7 | 46.7 | 41.3 | 43.0 | 46.0 |
| | InstructBLIP (13B) | 56.0 | 50.0 | 42.0 | 49.3 | 56.0 | 45.3 | 54.7 | 53.3 | 52.3 | 50.8 |
| | Video-LLaMA 2 (7B) | 19.0 | 13.0 | 18.0 | 16.7 | 18.7 | 15.7 | 10.7 | 20.0 | 21.7 | 19.2 |
| | Video-LLaMA 2 (13B) | 24.0 | 32.0 | 67.0 | 41.0 | 50.7 | 45.3 | 56.0 | 52.0 | 51.0 | 46.0 |

Table 6: Effect of finetuning baseline models on our training set. We compare the the finetuned Video-LLaMA 2, i.e., Video-LLaMA 2 (FT), and the pretrained Video-LLaMA 2. Note that the baselines here are multimodal models. So they are evaluated in the multimodal (MM) condition and the video-only condition but not in the text-only condition.

| | Method | Belief Inference | | | | Goal Inference | | | | | All |
|---|---|---|---|---|---|---|---|---|---|---|---|
| | | 1.1 | 1.2 | 1.3 | All | 2.1 | 2.2 | 2.3 | 2.4 | All | |
| Video MM | Video-LLaMA 2 | 36.0 | 38.0 | 52.0 | 42.0 | 41.3 | 36.0 | 30.7 | 45.3 | 38.3 | 40.2 |
| | Video-LLaMA 2 (FT) | 61.0 | 51.0 | 42.0 | 51.3 | 41.3 | 44.0 | 45.3 | 44.0 | 43.7 | 47.5 |
| | Video-LLaMA 2 | 24.0 | 32.0 | 67.0 | 41.0 | 45.3 | 50.7 | 56.0 | 52.0 | 51.0 | 46.0 |
| | Video-LLaMA 2 (FT) | 44.0 | 58.0 | 44.0 | 48.7 | 60.0 | 41.3 | 56.0 | 49.3 | 51.7 | 50.2 |

Table 7: Generalization results on the human test set. All models are the same as the ones evaluated in Table 1.

| | Method | Belief Inference | | | | Goal Inference | | | | | All |
|---|---|---|---|---|---|---|---|---|---|---|---|
| | | 1.1 | 1.2 | 1.3 | All | 2.1 | 2.2 | 2.3 | 2.4 | All | |
| Multimodal | InstructBLIP | 43.8 | 57.9 | 48.0 | 49.9 | 53.3 | 46.7 | 40.0 | 53.3 | 48.3 | 49.1 |
| | Video-LLaMA 2 | 50.0 | 47.4 | 24.0 | 40.5 | 33.3 | 33.3 | 20.0 | 46.7 | 33.3 | 36.9 |
| | GPT-4 with captions | 100 | 66.7 | 52.0 | 72.9 | 40.0 | 46.7 | 20.0 | 66.0 | 43.2 | 58.1 |
| | BIP-ALM w/ GPT-J | 93.8 | 52.6 | 88.0 | 78.1 | 73.3 | 86.7 | 73.3 | 66.7 | 75.0 | 76.6 |
| | BIP-ALM w/ LLaMA 2 | 87.5 | 73.7 | 96.0 | 85.7 | 60.0 | 86.7 | 60.0 | 66.7 | 68.3 | 77.0 |
| Text only | GPT-4 | 100 | 31.5 | 64.0 | 65.3 | 40.0 | 40.0 | 13.3 | 53.3 | 36.7 | 50.9 |
| | GPT-3.5 | 93.8 | 52.6 | 28.0 | 58.1 | 53.3 | 0.0 | 33.3 | 46.7 | 33.3 | 45.7 |
| | GPT-J | 62.5 | 42.1 | 44.0 | 49.5 | 33.3 | 53.3 | 26.7 | 53.3 | 41.7 | 45.6 |
| | LLaMA 2 | 75.0 | 52.6 | 44.0 | 57.2 | 46.7 | 40.0 | 60.0 | 53.3 | 50.0 | 53.6 |
| | BIP-ALM w/ GPT-J | 75.0 | 52.6 | 88.0 | 71.9 | 40.0 | 73.3 | 33.3 | 66.7 | 53.3 | 62.6 |
| | BIP-ALM w/ LLaMA 2 | 75.0 | 52.6 | 100 | 75.9 | 66.7 | 80.0 | 53.3 | 40.0 | 60.0 | 67.9 |
| Video only | InstructBLIP | 43.8 | 52.6 | 48.0 | 48.1 | 46.7 | 46.7 | 46.7 | 60.0 | 50.0 | 49.1 |
| | Video-LLaMA 2 | 31.3 | 26.8 | 68.0 | 42.0 | 53.3 | 60.0 | 40.0 | 40.0 | 48.3 | 45.2 |
| | GPT-4 with captions | 64.3 | 46.2 | 26.3 | 45.6 | 46.7 | 50.0 | 57.1 | 50.0 | 50.9 | 48.3 |
| | BIP-ALM w/ GPT-J | 62.5 | 57.9 | 60.0 | 60.1 | 46.7 | 53.3 | 73.3 | 60.0 | 58.3 | 59.2 |
| | BIP-ALM w/ LLaMA 2 | 75.0 | 57.9 | 76.0 | 69.6 | 60.0 | 53.3 | 60.0 | 53.3 | 56.7 | 63.1 |

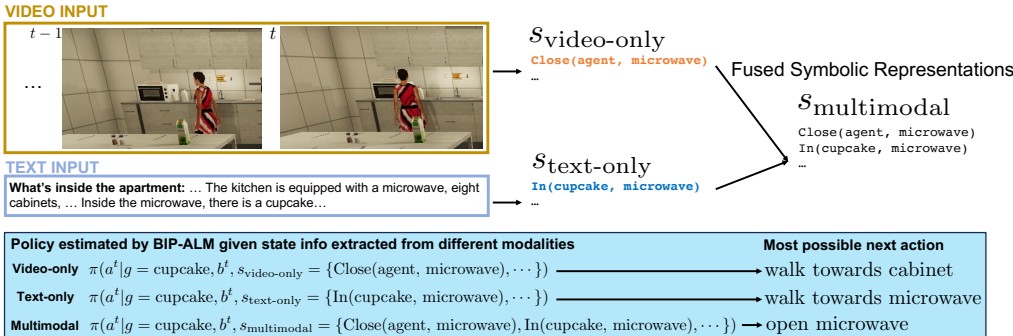

Figure 5: Example of how different modalities contribute to the mental state reasoning by BIP-ALM. In this example, we can get different state information from different modalities. From the video frame at step $t$, we can see that the person is close to the microwave, but we do not know what is inside the microwave or where we can find a cupcake. From the text, we know that there is a cupcake inside the microwave, but we do not know if the person is close to the microwave. By combining the two modalities, we can form a full picture of the world state. In the blue panel at the bottom, we show how state information from different modalities may shape the policy estimated by the model.

our model at these two steps reflect this reasoning. Consequently, our model answers the question correctly.

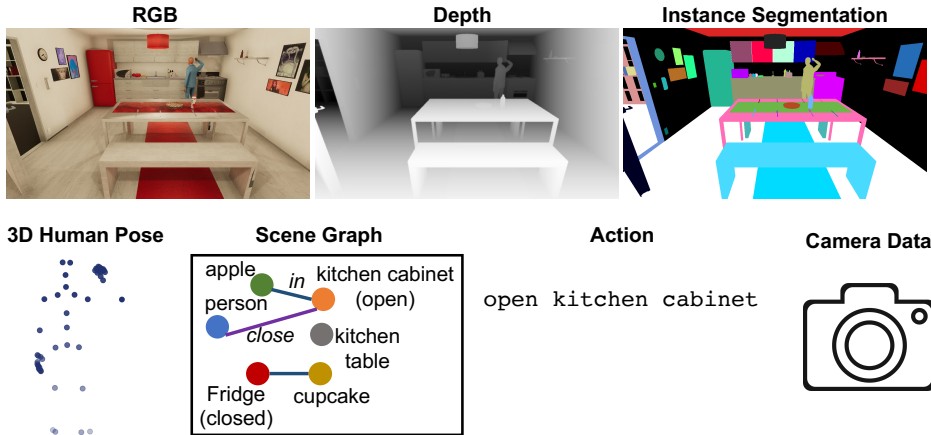

Figure 6: Types of data provided in our benchmark.

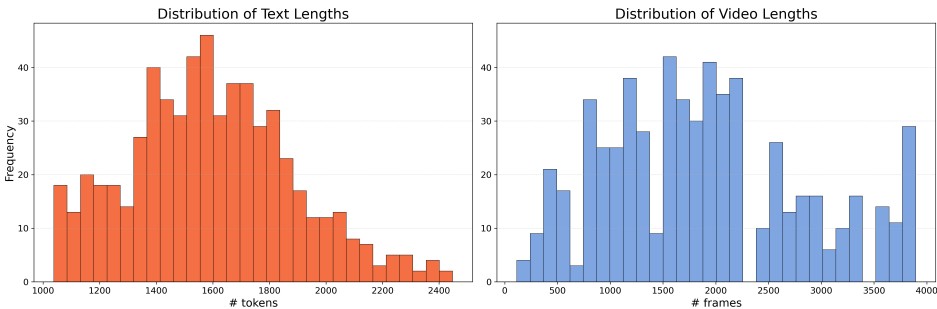

Figure 7: Context length of MMToM-QA.

Figure 5 illustrates how BIP-ALM may form different state information from different modalities and how different state information may shape the policy estimated by our model. In particular, from the video, we know that the person is close to the microwave but we do not know where the cupcake is. So the policy conditioned on the state extracted from the video ($S_{\text{video-only}}$) tries to guess where the person is going to look for the cupcake (e.g., cabinet). From the text, we know that there is a cupcake inside the microwave, but we do not know that the agent is already close to the microwave. Thus the policy conditioned on the state extracted from the text ($S_{\text{text-only}}$) predicts that the person is going to walk towards the microwave. After fusing information from both the video and the text, we then know the full state information. Conditioned on this fused state $S_{\text{multimodal}}$, the policy then predicts that the person is going to open the microwave, which is the ground-truth action of the person at the next step. This demonstrates that BIP-ALM can more accurately estimate the person's policy when having access to both modalities, which explains why it performs the best in the multimodal condition.

## B.3 AVAILABLE DATA

As shown in Figure 6, our benchmark provides RGB-D images, instance segmentation maps, human 3D poses, ground-truth scene graphs, ground-truth actions, and camera data.

## B.4 BENCHMARK STATISTICS

The environment in each question features an apartment that includes a bedroom, kitchen, living room, and bathroom. On average, each apartment contains 10.1 distinct types of containers (e.g., fridge, cabinets) and surfaces (e.g., desks, kitchen tables, coffee tables) types, and 16.4 instances

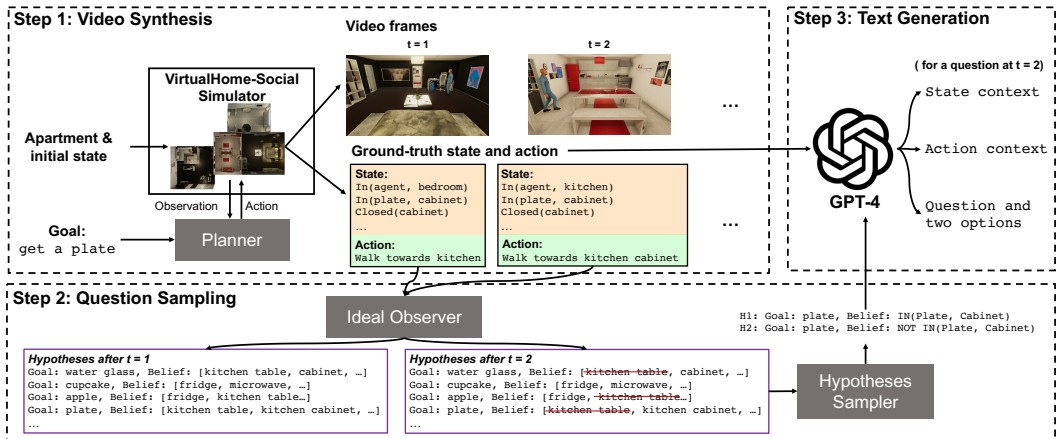

Figure 8: Overview of the procedural generation for creating our benchmark.

of containers and surfaces. There are on average 12 different object types, summing up to approximately 26.3 object instances in an apartment.

Figure 7 provides an overview of the distribution of text and video lengths across all questions in our benchmark. In comparison to existing ToM benchmarks, MMToM-QA features a more extensive text and video context, with an average of 1595 tokens and 1902 frames, respectively. Such lengths demand advanced information retrieval and fusion capabilities. The longer visual and textural context also increases the difficulty of paying attention to the relevant information to reconstruct the mental state of a person.

## B.5 DETAILS OF THE PROCEDURAL GENERATION

Figure 8 provides an overview of the procedural generation of questions in our benchmark. To procedurally generate videos, we first sample different apartments, the goal, and the initial state of an agent, then generate a sequence of actions using a planner. In particular, we formulate the agent in POMDP. The belief of the agent is represented as the probability of finding each object at a location. The agent can observe any objects within the same room that are not inside closed containers. We adopt the same planner in Puig et al. (2020), which has been verified to be able to synthesize human-like plans in household environments. Given the action sequence, we then render the RGB-D video frames and record ground-truth data including the instance segmentation maps, 3D human poses, scene graphs, actions, ground-truth agent beliefs, and the true goal.

To generate questions, we first use an "ideal observer" model to keep track of all possible goal and belief hypotheses at any given step. This model has access to the ground-truth observation of the agent at each step and eliminates the belief hypotheses that are inconsistent with the observations. It also has access to the planner of the agent and will evaluate each remaining goal and belief hypothesis pair by simulating actions conditioned on the hypothesis. If the simulated actions are consistent with the actual actions taken by the agent, we then further eliminate that hypothesis. Finally, to generate a question of a certain type at a given step, we sample two hypotheses that fit the design of the question type, with one from the set of possible hypotheses produced by the ideal observer model and the other being a hypothesis ruled out by the model.

At any given moment in a video, we ensure that our benchmark poses at most one question. To achieve this, we randomly select one question from the pool of possible questions at every step. We further sample a subset of the remaining questions to achieve a balanced distribution of question types.

### B.6 Utilizing GPT-4 for Enhanced Text Generation

We first translate the symbolic representations of state and action descriptions into natural language using simple templates. We then use GPT-4 to enhance the phrasing and diversify the expression. The prompts are provided as follows.

```
Improving state descriptions: We are describing where things are in an apartment. Please
improve the grammer of the description without changing the meaning. Please use only one
line break to separate descriptions about each room.
Original: There is a kitchen and a livingroom. 4 kitchencabinets are inside the kitchen.
There is nothing inside the 1st kitchencabinet from left to right. There is nothing inside
the 3rd kitchencabinet from left to right. There is nothing inside the 2nd kitchencabinet
from left to right. A waterglass and a wineglass and 2 dishbowls are inside the 4th
kitchencabinet from left to right.
Improved: There is a kitchen and a living room. The kitchen has four cabinets. The
first, second, and third cabinets, from left to right, are empty. The fourth cabinet, from
the left, contains a water glass, a wine glass and two dish bowls.
Original: {state description in the question}
Improved:
```

```
Improving action descriptions: Please improve the following descriptions about a person's
actions without changing the meaning.
Original: [name] is in the kitchen, walktowards stove.
Improved: [name] is in the kitchen. [name] walks towards the stove.
Original: [name] is in the livingroom, walktowards kitchen, walktowards 1st
kitchencabinet, open 1st kitchencabinet, close 1st kitchencabinet.
Improved: [name] is in the living room. [name] walks to the kitchen, approaches the first
cabinet, opens it, and then closes it.
Original: {action description in the question}
Improved:
```

## C  BIP-ALM Implementation Details

### C.1  Visual Perception

We adopt Blukis et al. (2022) to obtain a voxel map for each frame. Specifically, we first get the instance segmentation map from the RGB image. Combined with the depth map and the camera data, we create a 3D point cloud, with each point representing the pixel on the instance segmentation map. We group the 3D point cloud into a voxel map and then estimate the 3D bounding boxes of the objects. We can also estimate the 3D human pose in each frame. In the current experiments, we use ground-truth instance segmentation maps and ground-truth 3D human poses. In future work, we plan to also evaluate our model on segmentation and pose estimation results acquired from off-the-shelf computer vision models.

Using the 3D bounding boxes and the 3D human pose, we construct a scene graph. Each node in the graph is either an object or a person. For nodes representing containers, we indicate whether they are open or closed (which in our scenarios can be detected from the change in the sizes of their bounding boxes). There are two types of edges in a graph – (1) *inside* edges indicating containment and (2) *close* edges indicating proximity. Note that our BIP-ALM model is not restricted to these relationships and can be applied to broader scenarios with more types of spatial relationships.

### C.2  Text Parsing

Utilizing GPT-4, we parse the provided question to extract representations about the state context, action context, question, and the two options. In this subsection, we detail all the prompts used.

To extract representations regarding the state context, we use the following prompt:

```
State Extraction: Please extract the description of the rooms and where things are
in an apartment, found after the phrase ''What's inside the apartment'' and before the
description of a person's actions. Keep the line breaks.
Input: {question}
Extracted:

State Parsing: Please parse the following description about where things are in an
apartment. Each sentence should follow the pattern '[something] is/are in/on the
```

```
[somewhere].'  Use a '.'  to separate the sentences, and keep the original line breaks.

Original:  The living room contains a sofa, a desk, a cabinet, and a coffee table, and
the cabinet holds chips, a wine glass, and an apple.
Parsed:  A sofa, a desk, a cabinet and a coffeetable are in the livingroom.  Chips, a
wineglass and an apple are in the cabinet.

Original:  The kitchen has an oven, a microwave, and four cabinets.  The oven contains
a salmon, the microwave holds a cupcake, the third cabinet from the left has a wine glass,
the fourth cabinet is empty.  The first and second kitchen cabinets each holds a plate.
Parsed:  an oven and a microwave and 4 kitchencabinets are in the kitchen.  A salmon is
in the oven.  A cupcake is in the microwave.  A wineglass is in the 3rd kitchencabinet.
Nothing is in the 4th kitchencabinet.  A plate is in the 1st kitchencabinet.  A plate is in
the 2nd kitchencabinet.

Original:  {extracted states}
Parsed:
```

To parse the human actions, we employ the following prompt:

```
Action Extraction:  Please extract the exact description of a person's actions (starting
from the initial location), found after the phrase ''[someone]'s action'' and before the
question.  Please do not include the question, choices, or the answer.
Input: {question}
Extracted:

Action Parsing:  Please parse the description of a person's actions.  Use a '.'  to
separate each action, and remove all occurrences of the word 'and' in the description.

Original:  Jennifer is in the bedroom.  She proceeds to the kitchen and strides towards
the oven, preparing to open it.
Parsed:  In the bedroom.  walktowards kitchen.  walktowards oven.  about to open oven.

Original:  Mark is in the bathroom.  He then walks to the kitchen.  He sequentially
approaches the oven, the second, and third kitchen cabinets, opening and closing each
one in turn.
Parsed:  In the bedroom.  walktowards kitchen.  walktowards oven.  open oven.  close oven.
walktowards 2nd kitchencabinet.  open 2nd kitchencabinet.  close 2nd kitchencabinet.  open
3rd kitchencabinet.  close 3rd kitchencabinet.

Original:  {extracted actions}
Parsed:
```

To parse and analyze the question, we prompt GPT-4 to determine if a question falls under the "Belief Inference" or "Goal Inference" category, and extract all the hypothetical beliefs, hypothetical goals, and conditions in the question:

```
Question Parsing:    Please determine the type of inference for the input question:  either
''Belief Inference'', which inquires about a person's belief regarding an object, or ''Goal
Inference'', which seeks to understand a person's objective.
If a question falls under the ''Belief Inference'', please identify the [object] and the
[container] that the object may or may not be inside in choices (a) and (b).
If a question falls under the ''Goal Inference'', please identify the two possible objects
that the person is looking for in choices (a) and (b).  If the input contains a statement
indicating that someone believes there isn't an [object] inside a [container], please also
identify both the [object] and the [container] mentioned.  Otherwise, return 'NaN.'

Input:  ...  (detailed descriptions about states and actions) ...  If Elizabeth has been
trying to get a plate, which one of the following statements is more likely to be true?
(a) Elizabeth thinks that there is a plate inside the fridge.  (b) Elizabeth thinks that
there isn't any plate inside the fridge.
Output:  Belief Inference.  plate, fridge.

Input:  ...  (detailed descriptions about states and actions) ...  If Jennifer has been
trying to get a plate, which one of the following statements is more likely to be true?
(a) Jennifer thinks that there is a salmon inside the oven.  (b) Jennifer thinks that there
isn't any salmon inside the oven.
Output:  Belief Inference.  plate, fridge.  salmon, oven.

Input:  ...  (detailed descriptions about states and actions) ...  Which one of the
following statements is more likely to be true?  (a) Mark has been trying to get a plate.
(b) Mark has been trying to get a cupcake.
Output:  Goal Inference.  plate, cupcake.  NaN.

Input:  ...  (detailed descriptions about states and actions) ...  If Mary think there
isn't an apple inside the microwave, which one of the following statements is more likely
to be true?  (a) Mary has been trying to get an apple.  (b) Mary has been trying to get a
```

```
bottle of wine.
Output:  Goal Inference.  apple, wine.  apple, microwave.

Input:  {question}
Output:
```

## C.3  REPRESENTATION FUSION

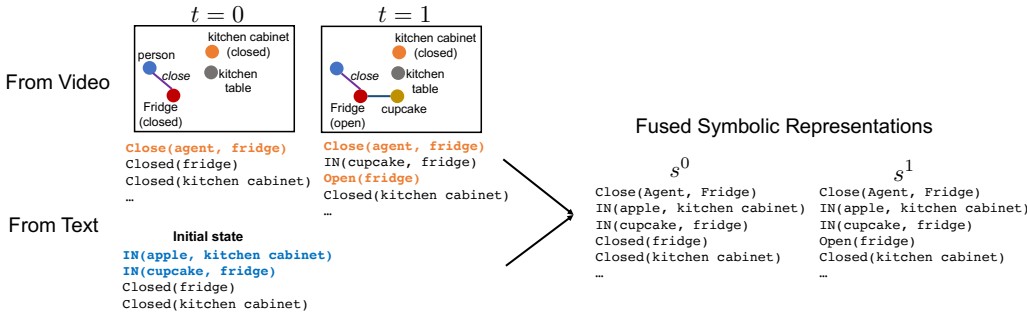

Figure 9: Illustration of fusing multimodal representations.

Figure 9 illustrates how BIP-ALM fuse complementary information extracted from the video and the text to form unified symbolic representations. In particular, the orange predicates can only be extracted from the video and the blue predicates can only be extracted from the text. After merging predicates extracted from both modalities, we can then construct a full state sequence.

When fusing information from different inputs, we may encounter conflicting predicates. In the case of conflict, we only keep the predicates from the text and remove the contradictory predicates from the video. This is because the predicates from the video are more likely to be erroneous due to noisy visual perception. However, more broadly speaking, such conflict-resolving mechanisms can be calibrated by the reliability of different modalities in a given domain.

## C.4  BELIEF REPRESENTATION AND UPDATE

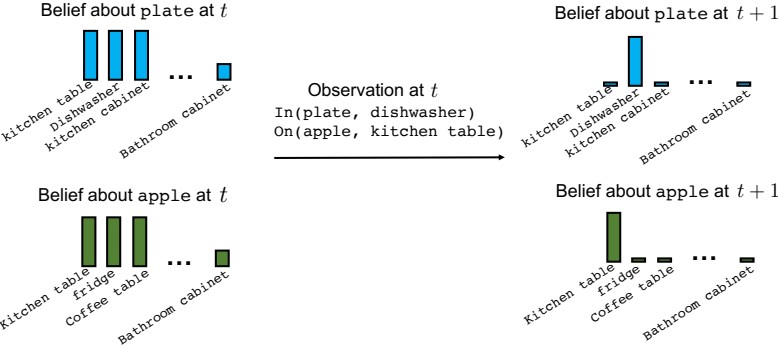

Figure 10: Illustration of estimated belief update.

A person's belief consists of beliefs about the locations of individual objects as illustrated in Figure 10. At each step, we estimate the person's observation and update the estimated belief accordingly $\hat{b}^t$. Then for each step, we construct a symbolic representation of the belief about each object as a list of possible locations of the object that have non-zero probabilities in the belief.

Note that for representing the negation of a predicate, we can simply remove the location from the list of possible locations in the symbolic belief representation.

### C.5 Prompt for Language Models in Inverse Symbolic Planner

As introduced in the main paper, we employ a language model (either GPT-J or LLaMA 2) to amortize the policy and estimate the likelihood of the person's last action $a^t$ given the hypothetical belief $b_t$ and goal $g$. We present the language model with the specified prompt below and then record its likelihood of predicting the accurate action.

```
goal: {hypothetical goal}
state: {state}
belief (possible locations the person suspects the {hypothetical goal} could be):
{hypothetical or predicted belief}
action:
```

### C.6 Training Details

In the experiments, we finetuned GPT-J (6B) and LLaMA 2 (7B) for our BIP-ALM method. Adhering to our evaluation protocol, we employed the ground-truth state, belief, goal, and action from 1,000 training videos, excluding the use of any example QAs. The models were trained using the state, belief, and goal as inputs at specific timestamps, with the aim of predicting the corresponding action. This approach enhanced the Inverse Symbolic Planner's ability to estimate the likelihood of human actions at a specific moment conditioned on the hypothesis about the goal and belief. The training data comprised 20,000 samples, consistent with the input and output of the Inverse Symbolic Planner defined in Appendix C.5.

For finetuning, we incorporated the Low-Rank Adapters (LoRA) method (Hu et al., 2021). The training process leveraged the AdamW optimizer, with a set learning of $5 \times 10^{-5}$ and a batch size of 4. We trained both models for 5 epochs, which took about 20 GPU hours on a single A100 GPU.

## D Implementation Details of Baselines

In all baselines, we use `gpt-4-0613` for GPT-4 and `text-davinci-003` for GPT-3.5.

In our **GPT-4 with captions** baseline, we use the following prompts to guide Video-LLaMA 2 to generate captions that summarize video content. These captions are subsequently integrated with our questions and provided to GPT-4 for question answering.

```
Descibing the Scenes in Videos:  Please describe what's inside the apartment based on the
video.  Please avoid using bullet points or numbers in your description.

Descibing Human Activity in Videos:  Please describe the actions of the human in the video.
Please avoid using bullet points or numbers in your description.
```

In particular, for the multimodal condition, we use the following prompt for **GPT-4 with captions** to generate the answer to a given question:

```
Captions:
What's inside the apartment:  {environment captions generated by Video-LLaMA}
The person's action:  {action captions generated by Video-LLaMA 2}
Text Descriptions:
{text description in the original question}
Question:  ...Please respond with either a or b, even when you are extremely uncertain
about the answer (Enforce GPT-4 to return a valid letter answer.)
```

In the video-only condition, we use the following prompt for **GPT-4 with captions** to generate the answer:

```
Captions:
What's inside the apartment:  {environment captions generated by Video-LLaMA}
The person's action:  {action captions generated by Video-LLaMA 2}
Question:  ...Please respond with either a or b, even when you are extremely uncertain
about the answer (Enforce GPT-4 to return a valid letter answer)
```

**Finetuning Video-LLaMA 2.** We followed Zhang et al. (2023) to create a video instruction dataset using our training data to finetune Video-LLaMA 2. In particular, based on our training videos

and the ground-truth annotations, we generated 7088 training examples for describing either the environment or the actions of the person in a given video clip. For each video clip, we sampled 8 frames and used the ground-truth scene graph at the last frame or the action sequence during the whole clip to generate the ground-truth descriptions for the training data. We then finetuned the Video-LLaMA 2 (13B) model on our training data on 2 A100 GPUs.

# E  FULL VERSION OF THE EXAMPLE QUESTIONS IN FIGURE 2

## E.1  TYPE 1.1 EXAMPLE

The video input: `https://youtu.be/4zoDwQk91ak`.

The text input:

```
What's inside the apartment:
The apartment consists of a bedroom, kitchen, living room, and bathroom.
In the bedroom, there is a coffee table and a desk, with a remote control resting on the
coffee table.
The kitchen is equipped with four cabinets, a fridge, a kitchen table, a microwave, and
an oven.  The first and third cabinets, from left to right, are empty, while the second
cabinet houses a condiment bottle.  The fourth cabinet contains a water glass.  Inside the
fridge, you'll find a bottle of wine, a dish bowl, and two plates.  The microwave holds a
cupcake, and the oven contains a salmon, two cupcakes, and a plate.
The living room features a cabinet, a sofa, a coffee table, and a desk.  The cabinet is
filled with two plates, a bottle of wine, two wine glasses, a condiment bottle, a water
glass, a bag of chips, and an apple.  A remote control, a wine glass, and a book are placed
on the coffee table.
Lastly, the bathroom has a cabinet, which is currently empty.
Actions taken by Elizabeth:
Elizabeth is initially in the bathroom.  She then proceeds to the kitchen and heads towards
the oven.  After opening and closing the oven, she moves to the second kitchen cabinet,
opens it, and then shuts it.  She repeats this action with the third and first kitchen
cabinets.  Subsequently, she walks towards the fourth kitchen cabinet, opens it, and then
closes it.  Finally, she moves towards the fridge, preparing to open it.

Question:
If Elizabeth has been trying to get a bottle of wine, which one of the following statements
is more likely to be true?
(a) Elizabeth thinks that there is a bottle of wine inside the fridge.
(b) Elizabeth thinks that there isn't any bottle of wine inside the fridge.
```

Correct answer: a.

## E.2  TYPE 1.2 EXAMPLE

The video input: `https://youtu.be/mgvh11Y_Z38`.

The text input:

```
What's inside the apartment:
The apartment consists of a bedroom, kitchen, living room, and bathroom.
In the bedroom, there is a coffee table and a desk, with three wine glasses and a dish bowl
placed on the coffee table.
The kitchen is equipped with four cabinets, a fridge, a kitchen table, a microwave, and
an oven.  The first, second, and fourth cabinets, from left to right, contain a dish bowl
each, while the third cabinet houses a plate.  The fridge contains two apples, a dish bowl,
and a salmon.  The microwave holds two cupcakes, and there is a salmon in the oven.
The living room features a cabinet, a sofa, a coffee table, and a desk.  The cabinet is
filled with a plate, a bag of chips, a water glass, a remote control, a bottle of wine, and
a condiment bottle.
Lastly, the bathroom has a cabinet, which is currently empty.
Actions taken by Jennifer:
Jennifer is situated in the living room.  She heads towards the cabinet and is about to
open it.

Question:
If Jennifer has been trying to get a cupcake, which one of the following statements is more
likely to be true?
(a) Jennifer thinks that there isn't any cupcake inside the cabinet.
```

> (b) Jennifer thinks that there is a cupcake inside the cabinet.

Correct answer: b.

### E.3 TYPE 1.3 EXAMPLE

The video input: `https://youtu.be/1N810N3KdjM`.

The text input:

> **What's inside the apartment:**
> The apartment consists of a bedroom, kitchen, living room, and bathroom.
> In the bedroom, there is a sofa with a book on it and a cabinet containing a remote
> control, a wine glass, two dish bowls, a bottle of wine, and a condiment bottle.
> The kitchen is equipped with a fridge, sofa, dishwasher, eight cabinets, an oven, a
> microwave, and a kitchen table. The fridge contains an apple, three plates, and a bottle
> of wine, while a bag of chips rests on the sofa. Inside the dishwasher, there is a plate,
> a water glass, and a wine glass. The first to the seventh cabinets, from left to right,
> are all empty. However, the eighth cabinet houses a wine glass. The oven contains a
> salmon, the microwave is empty, and the kitchen table is adorned with a plate, a wine
> glass, two apples, two books, and a cupcake.
> The living room features a sofa with a water glass and a book on it, and a desk.
> Lastly, the bathroom has a cabinet, which is currently empty.
> **Actions taken by Charles:**
> Charles is in the kitchen. He walks to the seventh kitchen cabinet, opens and closes it.
> He repeats the same action with the sixth kitchen cabinet. Subsequently, he moves towards
> the dishwasher.
>
> **Question:**
> If Charles has been trying to get a salmon, which one of the following statements is more
> likely to be true?
> (a) Charles thinks that there is a salmon inside the fridge.
> (b) Charles thinks that there isn't any salmon inside the fridge.

Correct answer: b.

### E.4 TYPE 2.1 EXAMPLE

The video input: `https://youtu.be/NsOPbJWPn1c`.

The text input:

> **What's inside the apartment:**
> The apartment consists of a bedroom, a bathroom, a living room, and a kitchen.
> In the bedroom, there is a coffee table with a plate on it.
> The bathroom houses a cabinet, which is currently empty.
> The living room is furnished with a cabinet, a coffee table, a sofa, and a desk. The
> cabinet is filled with two apples, a condiment bottle, three wine glasses, two water
> glasses, a cupcake, two bags of chips, a remote control, and a bottle of wine. Both a
> water glass and a wine glass are placed on the coffee table.
> The kitchen is equipped with a fridge, an oven, a kitchen table, and a microwave. Inside
> the fridge, there are two apples. The oven contains a salmon. Meanwhile, the microwave
> houses a salmon and two cupcakes.
> **Actions taken by James:**
> James is in the kitchen. He strides towards the stove, opens it, and then shuts it. He
> then opens the fridge, closes it, opens the microwave, and closes it as well. Finally, he
> walks towards the living room and approaches the cabinet.
>
> **Question:**
> Which one of the following statements is more likely to be true?
> (a) James has been trying to get a bottle of wine.
> (b) James has been trying to get an apple.

Correct answer: a.

### E.5 TYPE 2.2 EXAMPLE

The video input: `https://youtu.be/Fn6s47ZtxMQ`.

The text input:

```
What's inside the apartment:
The apartment consists of a bedroom, bathroom, living room, and kitchen.
In the bedroom, there is a sofa, a cabinet, a desk, and a coffee table.  A book rests on
the sofa.  The cabinet contains a remote control, three cupcakes, a wine glass, an apple,
and a bag of chips.  The coffee table holds two books and a dish bowl.
The bathroom houses a single cabinet, which is currently empty.
The living room is furnished with a sofa, a desk, and a coffee table.  A dish bowl and a
book are placed on the sofa, while a plate sits on the coffee table.
The kitchen is equipped with a dishwasher, an oven, a kitchen table, eight cabinets, a
microwave, and a fridge.  Inside the dishwasher, there is a dish bowl.  The oven contains
a salmon and a plate.  The second kitchen cabinet from the left holds an apple, while the
fourth and fifth cabinets contain two dish bowls and another apple respectively.  There is
a water glass inside the seventh cabinet.  The first, third, sixth, and eighth cabinets are
empty.  The microwave contains a condiment bottle.  The fridge stores two cupcakes, a dish
bowl, a plate, and a bottle of wine.
Actions taken by Mark:
Mark is in the kitchen.  He then advances towards the seventh kitchen cabinet.

Question:
If Mark thinks there isn't a water glass inside the 7th kitchen cabinet, which one of the
following statements is more likely to be true?
(a) Mark has been trying to get a water glass.
(b) Mark has been trying to get a cupcake.
```

Correct answer: b.

### E.6   TYPE 2.3 EXAMPLE

The video input: https://youtu.be/IUJW6Zv0EWA.

The text input:

```
What's inside the apartment:
The apartment consists of a bedroom, bathroom, living room, and kitchen.
In the bedroom, there is a cabinet and a sofa.  The cabinet contains a condiment bottle, an
apple, two wine glasses, and a plate.  The sofa holds three books.
The bathroom features a cabinet, which is currently empty.
The living room is furnished with a desk and a sofa, with a book resting on the sofa.
The kitchen is equipped with eight cabinets, a sofa, an oven, a fridge, a kitchen table, a
microwave, and a dishwasher.  The first kitchen cabinet, from left to right, contains a bag
of chips.  The second and fourth cabinets are empty.  The third cabinet houses a wine glass
and a dish bowl.  The seventh cabinet stores two plates.  The fifth, sixth, and eighth
cabinets are empty.  The oven contains a cupcake.  The fridge holds a plate and a dish
bowl.  The kitchen table is adorned with an apple, a bottle of wine, a plate, and a water
glass.  The microwave contains a condiment bottle and a salmon.  Lastly, the dishwasher has
a water glass inside.
Actions taken by Mary:
Mary is situated in the living room.  She proceeds towards the kitchen and heads to the
second kitchen cabinet.  She opens it, then promptly closes it.  She then opens the fourth
kitchen cabinet and closes it as well.  Following this, she opens the dishwasher and
closes it.  She then moves towards the sixth kitchen cabinet, opens it, and closes it.
She repeats this action with the seventh kitchen cabinet.  Finally, she walks towards the
first kitchen cabinet, opens it, and then closes it.

Question:
Which one of the following statements is more likely to be true?
(a) Mary has been trying to get a bag of chips.
(b) Mary has been trying to get a condiment bottle.
```

Correct answer: b.

### E.7   TYPE 2.4 EXAMPLE

The video input: https://youtu.be/Y4H_9cXR5mw.

The text input:

```
What's inside the apartment:
The apartment consists of a bedroom, bathroom, living room, and kitchen.
In the bedroom, there is a sofa, a cabinet, a desk, and a coffee table.  A book rests on
```

```
the sofa.  The cabinet houses an apple, a wine glass, two books, and two cupcakes.  The
coffee table holds a book, a water glass, a wine glass, and a remote control.
The bathroom contains a single cabinet, which is currently empty.
The living room is furnished with a sofa, a coffee table, and a desk.  A water glass sits
on the sofa, and a remote control is on the coffee table.  The kitchen is equipped with
eight cabinets, a microwave, a fridge, a dishwasher, a kitchen table, and an oven.  The
fourth and seventh cabinets from the left, as well as the eighth, are empty.  The microwave
contains a salmon, a cupcake, and a condiment bottle.  The fridge is stocked with two
bottles of wine, a cupcake, an apple, and two dish bowls.  The dishwasher holds a dish
bowl, a wine glass, and a plate.  The second cabinet from the left contains a water glass.
The first cabinet from the left holds a bag of chips and a wine glass.  The fifth cabinet
has an apple, and the third cabinet contains a condiment bottle.  The sixth cabinet is
empty.  Lastly, there is a salmon in the oven.
Actions taken by William:
William is situated in the kitchen.  He advances towards the first kitchen cabinet, opens
it, and then shuts it.  Finally, he moves towards the fifth kitchen cabinet.

Question:
Which one of the following statements is more likely to be true?
(a) William has been trying to get a wine glass.
(b) William has been trying to get a dish bowl.
```

Correct answer: b.