# OpenReview forum: "MMToM-QA: Multimodal Theory of Mind Question Answering"
_ICLR.cc/2024/Conference — Submitted to ICLR 2024_

### Official Review · Reviewer_jYXa · 2023-10-30

**Soundness:** 3 good
**Presentation:** 3 good
**Contribution:** 3 good
**Rating:** 6
**Confidence:** 4

**Summary:**

This paper addresses the problem of the theory of mind (ToM) by taking advantage of multi-modal information and the Bayesian inverse planning method. A new dataset is constructed with a focus on the scenarios where agents perform a sequence of household activities. The proposed method designs a collection of prompts and leverages a large language model for converting the multi-modal information (visual scene graph and textual description) into representations suitable for probabilistic inference. Experimental results demonstrate the effectiveness of the method in multiple settings.

**Strengths:**

+ It is an interesting question whether or not large language models are able to infer the internal state of the other agents. The paper presents a new multi-modal dataset, which can facilitate future research along the direction.

+ The proposed probabilistic model could potentially enable the understanding of how models aggregate information to address the problem of ToM, e.g., the relationships between observations and the dynamic of belief.

+ The paper proposes a principled approach for integrating multi-modal information into an unified representation, which shows promise in inferring the belief and goal.

**Weaknesses:**

- Despite leveraging a probabilistic planning method and showing considerable improvement, the paper pays little attention to investigating the model’s underlying mechanism. It would be more interesting to understand how the observed information influences the inference of belief, and which strategies are used by the models for tackling the challenges.

- The authors highlight that the proposed method solves the problem in a zero-shot manner, however, the appendix mentions that it is fine-tuned with ground truth annotations, please explain. In addition, I feel like a considerable portion of important details are compressed into the appendix, which makes the main paper difficult to understand.

- Constructing visual scene graphs is a challenging task, especially in naturalistic data of broader domains. How would the accuracy of predicted scene graphs affect the performance of the proposed method? With the increasing complexity of the visual scenes (e.g., from synthetic household environments to naturalistic scenarios in the wild), what are the potential ways of generalizing the method? Additionally, it seems that the paper does not mention which models are used for visual perception.

- Related to the aforementioned comment, the textual parsing and information fusion rely heavily on pretrained large language models (e.g., GPT-4) and a collection of carefully-tuned prompts (for household scenarios), it is relatively unclear whether or not such a paradigm is able to accommodate the diversity of more complicated environments.

- It appears that there is a significant variance in the performance of different types of problems, any idea why (e.g., GPT-4 fails drastically in setting 2.3)?

**Questions:**

(1) How would the proposed method benefit understanding behind the model’s inference of ToM?

(2) What are the challenges and potential solutions for generalizing the method toward broader domains?

(3) Is there any typical failure modes of the method, and why?

---

> ### Author Response · Authors · 2023-11-20
> **Author response (1/3)**
>
> Thank you for your detailed reviews and constructive comments. We respond to your questions and concerns as follows.
>
> ## How the observed information influences the inference of belief, and which strategies are used by the models for tackling the challenges.
> Thanks for the suggestion. We have revised Section 6 to include these discussions. We also summarize them here.
> 1. From a video, a model gets the dynamic state change at each step, including which containers are opened at a  given step, where the agent is, what objects the agent is walking toward, and what objects it is passing by at a given step. This provides information about the state change as well as the agent’s path, which is only available from the video. A model needs this information to determine the agent’s expected action plans given a hypothesis about the belief and the goal.
> 2. From the text, a model gets ground truth information about the initial state, including all objects’ initial locations as well as the action label at each time step. Because of the partial observations caused by the limited camera view, the closed containers, and occlusion, the text provides additional information about where objects are initially, and the actions taken by the agent, which is sometimes not observable in the video. A model requires information about the true world state to determine whether an agent has a true belief or false belief as well as what objects the agent has observed so far. A model also needs the complete action sequence to judge whether a hypothesis about the belief and goal is consistent with the agent’s past actions.
>
> Our model successfully utilizes relevant information extracted from different modalities for reasoning about a person’s behavior conditioned on the belief and goal as discussed above. On the other hand, baselines fail to do so and consequently perform poorly.
>
> ## Zero-shot evaluation.
> Please refer to the general response to the zero-shot evaluation question.
>
> ## How would the accuracy of predicted scene graphs affect the performance of the proposed method?
> The scene graph construction method we used is very accurate given observable visual information (the predicted spatial correlations have a 95% accuracy). However, because of occlusion and limited camera view, there are parts of the environments we cannot clearly see from the video, which caused low overall accuracy in reconstructing the full world state. This is inevitable due to the nature of visual inputs. Human performance is lower in the video-only condition, which is also caused by the partial visual observation in the videos. However, humans sometimes can imagine what people might be doing or what objects could be there in the case of occlusion using their common sense, so they are more resilient to missing visual information. A challenge for future work could be developing a scene graph construction method to also imagine the missing visual information. We discussed this in the revised Section 6.
>
> ## What are the potential ways of generalizing the method?
> We acknowledged that we have not investigated how the method would perform in scenarios other than household environments. However, we believe that our work as a first step toward multimodal Theory of Mind could provide valuable guidance for future work to further pursue robust multimodal Theory of Mind in the wild. In particular, the following findings from the current work indicate several promising ways in which we can scale up the method:
> 1. In our ablated study, we found that our model using pretrained GPT-J or LLaMA without finetuning performed significantly better than using the original GPT-J and LLaMA alone; it even could outperform GPT-4 which is a much larger model. In the future, if we train larger LMs on larger human activity datasets collected in the wild, we can potentially scale up our method by using these better LMs.
> 2. Our current work demonstrates that synthetic human data created in embodied environments can be used to finetune LMs to better perform ToM reasoning. In the new experiments we conducted for the revision, we also showed that the finetuned models can reason about real humans’ actions and mental state updates (i.e., results on the human test set). There have been recent works on embodied simulators (e.g., the recently released Habitat v3.0) that support the synthesis of more realistic and complex human activities. So future simulators could provide training data for a broader range of human activity training data. We can apply the same method with LMs finetuned on these larger and better synthetic datasets as well to reason about real-world human behaviors.
>
> We have revised Section 5 and Section 6 to include these discussions.
>
> ## Which models are used for visual perception?
> In the revision, we further clarified that we used the method in Blukis et al. (2022) to generate the scene graph (the first paragraph in Section 4.1).

---

> ### Author Response · Authors · 2023-11-20
> **Author response (2/3)**
>
> ## It is relatively unclear whether or not such a paradigm is able to accommodate the diversity of more complicated environments.
>
> We want to clarify that our prompts are quite generic and we have already demonstrated the generalizability of the current textual parsing. In particular, we have not turned the prompt, but rather just designed a simple template that prompts GPT-4 to parse any given text to state predicates. The examples provided in the prompt only include about 5% of objects, actions, and state predicates. In the test set, there are many unseen objects, actions, and state predicates. However, we didn’t observe any failure in textual parsing. This demonstrates the generalization ability of the parsing method. We discussed how we may further apply the method to even more diverse environments in the revised Section 6.
>
> ## Why is there a significant variance in the performance of different types of problems?
> 1. GPT-4 excels at the basic belief inference question type (1.1), in which the person’s belief is the same as the true world state (true belief). However, it tends to confuse a person’s belief with the true world state and thus predicts the person’s belief to be the same as the true world state in 1.2 even though the person has a false belief (the belief is different from the true world state) in 1.2. It performs relatively well in 1.3 because the person always has a true belief in those questions. Its accuracy in 1.3 is not as high as 1.1, because 1.3 requires GPT-4 to keep track of the person’s belief over a long period and not just based on the latest action. This shows that GPT-4 has a limit in tracking a person’s belief over a long sequence of actions.
> 2. GPT-4 in general performs at the chance level for all goal types except 2.3 (which is lower than chance). In 2.3, the person updates the belief after checking inside a container and decides. Since they cannot find the goal object, they then close the container without picking any object from the container. GPT-4 consistently predicts that the person’s goal is to get one of the objects inside in 2.3 instead of predicting that the person is more likely to pursue objects that are not inside the container. We hypothesize that this is because GPT-4 fails to update the person’s belief based on the actions of opening and closing the container, and mistakenly thinks that the person opens the container because the goal has to be one of the objects inside the container.
>
> We have included these discussions in the revised Section 5.2 and Section 6.
>
> ## How would the proposed method benefit understanding behind the model’s inference of ToM?
> Instead of directly mapping the visual and language information to belief and goal predictions to generate the answer (i.e., how LLM and LMM baselines answer the questions), our proposed method conducts model-based inference (Bayesian inverse planning), by imaging possible actions given a hypothetical mental state. Such model-based inference captures the nature of how humans’ mental state decides humans’ actions, as studied in cognitive science, e.g., (e.g., Baker et al., 2009; Ullman et al., 2009; Baker et al., 2017; Shu et al., 2020).
>
> We have included this in the revised Section 6.
>
> ## What are the challenges and potential solutions for generalizing the method toward broader domains?
> We discussed some aspects of the generalization of the method in previous questions. Here, we summarize more comprehensively as follows:
> 1. There could be more types of human activities. The promise of finetuning LMs on larger datasets and/or using larger pretrained LMs is that these LMs can generate plans to estimate action likelihoods for Bayesian inverse planning in broader activities. We can also query an LLM to generate possible goals a person might want to pursue in the described environment. For instance, we can ask LLM what things a person may want to do in a house, in an office building, in a restaurant, at a traffic intersection, etc.
> 2. Scene graph construction in the wild is a challenging problem. However, there have been recent works on open vocabulary scene graph generation (e.g., ConceptGraphs proposed by Gu et al., 2023), which could be used to improve the generalizability of the visual perception module. The modular design of our method allows us to take advantage of the improved visual perception module without changing other components.
> 3. The rendered video frames may not be 100% realistic. However, the scene graph representation itself is a quite general representation that can be used in the wild. By training LMs to reason about a person’s possible actions given hypothetical mental states on top of the scene graph representation, we can avoid pixel-level domain gaps (i.e., rendered frames vs. real-world frames). Once we construct scene graphs from real-world video frames, the symbolic representations that LMs operate on are the same as the ones from the synthetic data.

---

> ### Author Response · Authors · 2023-11-20
> **Author response (3/3)**
>
> ## Are there any typical failure modes of the method, and why?
> Thank you for the suggestion. We revised Section 5.3 and Section 6 to further discuss the failure models of the method. We summarize them as follows as well:
> 1. The current method is unable to imagine missing state information from videos. As discussed in the earlier question about the effect of visual perception on the model performance, missing state information resulting from limited camera view or occlusion contributed to the low performance of the model in video-only scenarios.
> 2. The current model does not have motion planning, which is crucial for answering 2.4 as the model has to imagine possible future objects that a person is going to check given the person’s current path.
> 3. The action likelihood estimated by the LMs sometimes could be inaccurate due to LMs’ occasional failures in task planning.

---

> > ### Comment · Reviewer_jYXa · 2023-11-21
> >
> > I thank the authors for the detailed response, which addresses most of my concerns. For interpretation of the model's inference procedure, perhaps it will be better to include some qualitative examples for the dynamics of states/actions. In addition, it would be interesting to see how altering different components in the intermediate states would impact the model behaviors.

---

> > > ### Author Response · Authors · 2023-11-23
> > > **We added qualitative examples in the latest revision**
> > >
> > > Thanks for your suggestion. We have revised the paper to include a few qualitative examples in Figure 4 and Figure 5 (in the appendix) and discussed them in Appendix B.2. In Figure 4, we show how our model estimates the likelihood of mental state hypotheses by reasoning about the actions of a person given the state. It demonstrates why it is necessary to imagine the action policy of a person given the hypothesized mental state as well as the state in order to successfully conduct ToM reasoning. Figure 5 illustrates how different modalities contribute to the mental state reasoning by our model. It shows how altering the state information available to our model may change the action policy estimated by our model.

---

### Official Review · Reviewer_Ewvn · 2023-10-30

**Soundness:** 1 poor
**Presentation:** 2 fair
**Contribution:** 1 poor
**Rating:** 3
**Confidence:** 4

**Summary:**

The paper presents a multimodal dataset called MMToM-QA for evaluating machines' understanding of a person's activities in a household environment. Based on the dataset, the authors propose a BIP-ALM method to encode information from multimodal inputs and utilize language models (LMs) for inference. They find that LMs and multimodal models are still lacking the knowledge to solve MMToM-QA, but BIP-ALM shows a promising direction.

**Strengths:**

The paper is easy to follow and every details are clearly described.

**Weaknesses:**

MMToM-QA is flawed. VQA has been studied for years and the issues in crowdsourcing and annotations are clearly mentioned in many piror works. However, the paper lacks quantitative studies regarding these issues. For example, whether the questions designed can be attacked by counting the repeating semantic meaningful words, whether the text question already implied the answers. From Table 1, the high performance in text-only models may mean these flaws in dataset design.

The conclusion that "large language models and large multimodal models still lack robust ToM capacity" is not convincing, as the paper only tests BIP-ALM on the proposed MMToM-QA dataset. There are many other datasets that evaluate models' understanding of people's activities, such as VQA [1] and VCR [2]. The paper should compare BIP-ALM to the state-of-the-art methods on these benchmarks and also cite these benchmarks in related works.

Comparison to baselines is unfair. The paper tested BIP-ALM tuned on the dataset, yet use zero-shot approach to test the baselines such as GPT, LLaMA, InstructBLIP etc. So the pretraining of these baselines may fail to generalize to a totally different MMToM-QA data. The authors need to finetune these LLMs or prompt these models to provide a more fair comparison.

[1] "Vqa: Visual question answering."
[2] "From recognition to cognition: Visual commonsense reasoning."

**Questions:**

Please read the weaknesses I listed.

---

> ### Author Response · Authors · 2023-11-20
> **Author response**
>
> Thank you for your reviews and suggestions. We respond to your concerns and questions below.
>
> ## VQA has been studied for years and the issues in crowdsourcing and annotations are clearly mentioned in many prior works.
>
> We want to respectfully point out that our benchmark is evaluating a fundamentally different problem than VQA. We also did not use any crowdsourcing and annotations when constructing the dataset. We further clarified this in the revised related work section. We also summarize three key differences between MMToM-QA and VQA as follows.
> 1. Our benchmark evaluates whether a model can **infer a person’s hidden mental state (beliefs and goals)** from observable information about the world state and actions (as explained in the introduction). This is different from questions tested in VQA. Questions regarding human activities in VQA do not evaluate Theory of Mind.
> 2. Prior works (e.g., Netanyahu et al., 2021) have also shown that models that perform well in human activity recognition do not perform well in Theory of Mind reasoning. This further provides empirical evidence about the difference between ToM and human activity recognition.
> 3. Finally, our benchmark emphasizes the ability to extract useful information from **multimodal inputs**, which is not evaluated in VQA. In sum, our multimodal Theory of Mind benchmark studies a new problem that is fundamentally different from VQA benchmarks, both in terms of the evaluated model capacities, available modalities, and the construction of the benchmark. We further discuss this critical difference between our multimodal Theory of Mind in the revised related work section.
>
> ## From Table 1, the high performance in text-only models may mean these flaws in dataset design.
>
> As shown in the examples in Figure 2, each question in our benchmark features a unique environment and human actions; scenarios between different types of questions are also drastically different. There are no simple patterns a model can use to guess the answers and achieve high accuracy in all types of questions. In our benchmark, some models perform better in the text-only condition compared to the video-only condition, because there is more state information provided in the text and the textual parsing is also easier than constructing scene graphs from videos. For instance, from the video, we cannot see what’s inside a container when it is closed or objects occluded by the person. However, we can know this crucial information from the text.  Additionally, in our human experiments, human participants also performed better in the text-only condition compared to the video-only condition, which further demonstrates that the results are not caused by adversarial attacks or simple word patterns, but by the nature of the ToM tasks and how they rely on different kinds of modalities. Findings on how humans and models perform in different conditions and question types shed light on what kinds of information are necessary when reasoning about different aspects of the mental state, which we discussed in the revised Section 6.
>
> ## The conclusion that "large language models and large multimodal models still lack robust ToM capacity" is not convincing… There are many other datasets that evaluate models' understanding of people's activities.
>
> In the revised Section 5 and Section 6, we discussed the failures exhibited by large language models and large multimodal models, explaining why they lack robust ToM capacity. Since we are evaluating Theory of Mind reasoning, which is entirely missing in VQA and VCR, we believe that VQA and VCR are not suitable for our goal. Our benchmark is complementary to existing benchmarks by focusing on a model’s ability to infer a person’s mental states (goals and beliefs) from observable actions. We have cited and discussed these benchmarks in the related work.
>
> ## Zero-shot evaluation
>
> Please see our general response on the zero-shot evaluation. Briefly, a model (including ours) never sees any example QAs. This is following the standard evaluation protocol in existing ToM benchmarks (e.g., Shapira et al., 2023). A model can be trained on the training set, but the training set has no example QAs. Hence, even when a model is trained on the training set, it is still evaluated in a zero-shot manner. Our ablated study (Table 3 in Appendix B.1) shows that our method without training also outperforms baselines. In the revision, we ran additional experiments to evaluate few-shot and finetuning baselines. Please refer to our general response, Section 5.3, and Table 5 and 6 in Appendix B.1 for more details. We also further clarified the evaluation protocol in the revised Section 3.4, including the purpose of training.

---

### Official Review · Reviewer_pPZm · 2023-11-01

**Soundness:** 2 fair
**Presentation:** 2 fair
**Contribution:** 2 fair
**Rating:** 5
**Confidence:** 2

**Summary:**

The paper proposes a benchmark and a model for evaluating theory of mind in machine learning models. The benchmark consists of synthetic videos, explanations, questions, and answers. The associated proposed model leverages GPT-4 for translating natural language text into symbolic representations as well as a visual perception model to extract visual information into symbolic representations. Several baselines are evaluated on the proposed benchmark on three modalities: multimodal, text-only, and video-only.

**Strengths:**

The paper includes a multimodal QA dataset with synthetic videos that can be potentially useful for future research and includes a human baseline.

**Weaknesses:**

1. The paper should do a better job of justifying the proposed task itself. The paper argues that Theory of Mind is important for developing machines with human-level social intelligence. However, it is unclear why having machines with “human-level social intelligence” is necessary. First, can “social intelligence” be measured or evaluated? It is still unclear how to define intelligence alone and how to measure it within the machine-learning context. Second, what types of applications will benefit from this skill?

2. Without a clear goal and definition, it is hard to validate whether the proposed benchmark aligns with the task described in the introduction of the paper. Can we draw conclusions from the results in the proposed benchmark about a model’s social intelligence? It is hard to say in the current setting.

3. The text is unclear about using synthetic videos in the test set. As it is written in Section 3.1, it looks like the 134 test videos are real image videos, whereas the 1000 training videos are synthetically generated. However, in Section 3.3, it looks like the test videos are also synthetic. If test videos are not synthetic, how are they obtained?

**Questions:**

- What are the statistics of the benchmark per question type?
- Please include an overall column in Table 1 or the actual numbers in Figure 4.

---

> ### Author Response · Authors · 2023-11-20
> **Author response**
>
> Thank you for your reviews and helpful suggestions. We address your concerns and questions as follows.
>
> ## Justify the proposed task
>
> We further clarified the importance of social intelligence in the revised introduction. We also explained that we are focusing on one specific aspect of social intelligence, Theory of Mind (ToM), defined what Theory of Mind is, and discussed how it has been quantitatively evaluated in prior works. We also summarize it here:
>
> 1. Social intelligence is one of the fundamental aspects of human intelligence as cognitive science studies have pointed out (e.g., Wimmer & Perner, 1983; Gergely et al., 1995; Gergely & Csibra, 2003; Spelke & Kinzler, 2007). It serves as the foundation of human social interactions (Warneken & Tomasello, 2006) and social learning (Csibra & Gergely, 2006).
> 2. Theory of Mind is a key aspect of social intelligence, which has been demonstrated to be important for building socially intelligent systems, such as assistive robots (e.g., Dautenhahn et al. 2007, Hadfield-Menell et al., 2016; Puig et al., 2023), AI teachers (e.g., Wang et al., 2021), autonomous vehicles (e.g., Chandra et al., 2020), and cooperative embodied agents (e.g., Bara et al., 2021; Sclar et al., 2022a). These are well-studied in ML/AI, robotics, and HCI literature. **Instead of focusing on the broad concept of social intelligence, our work focuses on Theory of Mind.**
> 3. There have been many studies on design experiments to quantitatively evaluate humans’ Theory of Mind in cognitive science (e.g., Gergely & Csibra, 2003; Baker et al., 2009; Ullman et al., 2009; Baker at al., 2017; Shu et al., 2020) and on developing benchmarks to **quantitatively** evaluate machine Theory of Mind in ML/AL (Rabinowitz et al., 2018; Kosinski, 2023; Sap et al., 2019; 2022; Ullman, 2023; Shapira et al., 2023; Shu et al., 2021; Moghaddam & Honey, 2023; Nematzadeh et al., 2018; Gandhi et al., 2021). Specifically, the experiments and benchmarks in the prior works evaluate the accuracy of goal inference and belief inference based on observed behaviors and use these quantitative evaluations as an indicator of whether a model has ToM capacity. Our benchmark builds on these prior works with significant extensions including more complex scenarios and multimodal inputs.
>
> ## Goal and definition? Can we draw conclusions from the results in the proposed benchmark about a model’s social intelligence?
>
> In our introduction, we have stated the goal of the benchmark, which is to evaluate whether a model can infer the mental state of a person, including goals and beliefs, from multimodal inputs of that person’s behavior (i.e., multimodal Theory of Mind). We have defined what Theoy of Mind is (i.e., the ability to infer a person’s mental state (belief and goal) based on the observed behaviors) and discussed how prior works have been evaluating Theory of Mind and building models to engineer machine Theory of Mind. Our benchmark quantitatively evaluates a model’s belief and goal inference accuracy of a model in different scenarios. It also evaluates the effect of different modalities on human ToM and machine ToM. It is well grounded in the prior works on Theory of Mind in both cognitive science and ML/AI and significantly extends the prior works.  As discussed in Section 5 and Section 6, the successes and failures of a model in different types of questions reveal their capacities as well as limitations in Theory of Mind.
>
> ## The text is unclear about using synthetic videos in the test set.
> All videos in both the test set and training set are synthetic and are procedurally generated in the same way. We made this clearer in the revised Section 3.3.
>
> ## What are the statistics of the benchmark per question type?
> We added the number of questions for each question type in the revised Section 3.1.
>
> ## Please include an overall column in Table 1.
> Thanks for the suggestion. We added the overall column in the revised Table 1.

---

### Official Review · Reviewer_Djbm · 2023-11-01

**Soundness:** 2 fair
**Presentation:** 2 fair
**Contribution:** 2 fair
**Rating:** 5
**Confidence:** 4

**Summary:**

This benchmark evaluates ToM on multimodal data and various unimodal data, addressing the limitations of existing unimodal ToM benchmarks. The proposed method, BIP-ALM (Bayesian Inverse Planning Accelerated by Language Models), successfully integrates unified representations from multimodal data with scalable Bayesian inverse planning using language models. Comparative experiments reveal that while large language and multimodal models lack robust ToM capacity, BIP-ALM exhibits promising results by combining model-based mental inference and language models.

**Strengths:**

1. The motivation is clear and the problem tackled is indeed interesting and important.
2. The dataset contribution is helpful for the community.
3. The paper is overall well presented.

**Weaknesses:**

1. Currently, this benchmark still significantly lacks of comprehensive baseline comparisons and in-depth analysis to really show the audience **why** the identified problem is so important and challenging.

a. More baselines like few-shot GPT4/3.5, evaluating LLM/Multimodal LLM with chain-of-thought types of reasoning process, open-sourced models with different sizes should be further studied to provide a better understanding of the performance of existing methods on this task.

b. Error analysis of existing models like GPT-4 or VideoLLAMA should be provided since it is important for the audience to understand how the proposed method solves the flaws in existing methods exactly. With the recent release of GPT4V, it will be great if some insights could be drawn from this case study as well.

c. Currently the evaluation setting among different models is also not very clear. For example, models like VideoLLAMA actually takes video frames as input but other LLM models seem to only take parsed information. It is important to clearly annotate the exact format of each modality for all the model variants to make it clear to understand the possible difference from the input side.

2. Currently, some limitation/ design choice is not well justified.

a. Why is the proposed method not applied to VideoLLAMA and Instruct BLip?

b. Is the synthetic video data really capturing important goal and belief in real world? What is the domain gap? There should be at least case studies on some real procedural videos, investigating the possible domain shift to validate the usage of synthetic video and revealing the possible limitation of this benchmark more explicitly.

**Questions:**

Please check weakness for details.

**Details Of Ethics Concerns:**

Despite the usage of synthetic data, the authors should still further show that the synthesized data doesn't contain certain bias of gender or race, or any sensitive properties.

---

> ### Author Response · Authors · 2023-11-20
> **Author response (1/2)**
>
> Thank you for your helpful and constructive reviews. We respond to your concerns and questions below.
>
> ## More baselines
>
> We did not evaluate few-shot GPT 4/3.5 because we followed the standard evaluation in the prior ToM QA evaluation (e.g., Shapira et al., 2023), which adopted zero-shot evaluation. We provided more explanation in the general response and clarified this in the revised Section 3.4. Per your suggestion, in the revision, we evaluated few-shot GPT 4/3.5, LLM/LMM with chain-of-thought, and open-sourced models with different sizes. We have discussed the results in Section 5.3. Specifically,
>
> 1. We found no meaningful improvement for almost all baselines after using different few-shot or chain-of-thought prompting. They still perform no better than chance. The only exception is GPT-4 in the text-only condition, which has an improvement in simpler types (e.g., Type 1.3) with few-shot prompting but still performs poorly on harder types, e.g., false belief (Type 1.2).
> 2. We also found that all baselines performed no better than chance, regardless of the model sizes.
>
> ## Error analysis of existing models.
>
> Thanks for the suggestion. We have revised Section 5 and Section 6 to include more error analyses. We also summarize them here:
>
> 1. GPT-4 excels at the basic belief inference question type (1.1), in which the person’s belief is the same as the true world state (true belief). However, it tends to confuse a person’s belief with the true world state and thus predict the person’s belief to be the same as the true world state in 1.2 even though the person always has a false belief (the belief is different from the true world state) in 1.2. It performs relatively well in 1.3 because the person always has a true belief in those questions. Its accuracy in 1.3 is not as high as 1.1, because 1.3 requires the GPT-4 to keep track of the agent’s person over a long period and not just based on its latest action, which shows that GPT-4 has a limit in tracking a person’s belief over a long sequence of actions.
> 2. GPT-4 in general performs at the chance level for all goal types except 2.3. In 2.3, the person updates the belief after checking inside a container and decides. Since they cannot find the goal object, they then close the container without picking any object from the container. GPT-4 consistently predicts that the person’s goal is to get one of the objects inside in 2.3 instead of predicting that the person is more likely to pursue objects that are not inside the container. We hypothesize that this is because GPT-4 fails to update the person’s belief based on the opens and closes container, and mistakenly thinks that the person opens the container because the goal has to be one of the objects inside the container.
> 3. Large multimodal models, including Video-LLaMA and InstructBLIP fail to perform significantly better than chance for all types of questions, unlike GPT-4 which could excel in simple types. Additionally, we observe that captions generated by Video-LLaMA did provide useful information extracted from the visual inputs as the performance of captions + GPT-4 is better than GPT-4 alone as well as using Video-LLaMA to directly answer the questions. This demonstrates that while Video-LLaMA could provide useful captions describing
>
> ## The exact format of each modality for all the model variants.
>
> Only multimodal models (VideoLLaMA and InstructBLIP) can take video frames as inputs. Language models (GPT-4/3.5, GPT-J, LLaMA) can only take in parsed information. We clarified this in Section 5.1.
>
> ## Why is the proposed method not applied to VideoLLaMA and InstructBLIP?
>
> Our method is Bayesian Inverse Planning Accelerated by **Language Models**. We use language models to evaluate action likelihoods based on parsed symbolic information. This allows us to fuse information from both visual inputs and language inputs into a unified representation. It also allows us to finetune language models only on symbolic representations and apply the finetuned language models to any type of modalities. In contrast, VideoLLaMA and InstructBLIP are not language models but multimodal models that directly take in both video frames and language inputs. They are not what our method relies on.

---

> ### Author Response · Authors · 2023-11-20
> **Author response (2/2)**
>
> ## Case studies on some real procedural videos
>
> Thanks for the suggestion.
> 1) We used the method proposed in the prior work (Puig et al., 2020) to synthesize people’s mental states and actions since the prior work has demonstrated that there is no significant difference between synthetic behaviors and real human behaviors. Cognitive science studies (e.g., Baker et al., 2017) have also verified that human mental update and action planning can be captured by POMDP, which is how we define the simulated agent in our work.
> 2) To evaluate whether a model can understand real human mental states based on real human actions, we have additionally collected real human actions by using the human interface in the VirutalHome-Social platform. We then rendered videos based on collected **real human actions**. In particular, to create a human-generated video, we assigned a goal to a participant and asked the participant to control the avatar based on the egocentric observation to reach the goal. The resulting human actions are then used to render the video. We recruited 3 participants who had no prior exposure to the system. In total, we created 40 human-generated videos and created 120 questions as the human test set. The mental state updates in these human-generated videos are the real human mental state updates (human participants had to update their own beliefs to control the avatar to reach the assigned goal). Moreover, we used 2 new apartments that are unseen in the original test set and the training set in these human videos, so that we can further evaluate the generalizability of our method in unseen environments. Results are reported in Section 5.3 and Table 7 in Appendix B.1. It shows that our method performs at the same level in the human test set compared to the main test set. It demonstrates that our method can successfully generalize to real human behavior in unseen environments. It also suggests that synthetic behavior captures how humans update mental states and take actions.
> We leave the investigation of real human activities in broader goals for future work. Compared to prior ToM benchmarks which are all generated in toy domains, our benchmark has already taken a significant step toward ToM in realistic and complex scenarios.
>
> ## Further show that the synthesized data doesn't contain certain bias of gender or race, or any sensitive properties.
>
> We have made sure the equal gender representation in our data. We have also featured different races in the avatars. Moreover, we anonymized all data collected in human experiments. We are not aware of any sensitive properties.

---

### Author Response · Authors · 2023-11-20
**General response**

# General response

We thank reviewers for their constructive comments and for acknowledging that (1) the problem of multimodal ToM is interesting and important; (2) our motivation is clear; (3) the dataset can facilitate future research; (4) the paper is well presented; (5) we propose a principled approach which shows promising results; and (6) we have provided a human baseline.

We have substantially revised the paper (highlighted in blue) according to the reviews. We have also conducted the following experiments suggested by the reviewers:

1. We evaluated the open-source models with different model sizes.
2. We evaluated the effect of few-shot and chain-of-thought prompting for baselines.
3. We finetuned VideoLLaMA v2 on our training set.
4. We collected real human behaviors in unseen environments and created a human test set to evaluate the generalizability of the models.

We summarized and discussed the results of these experiments in Section 5.3 and Table 4 - 7 in Appendix B.1. Additionally, we have added a few qualitative results (Appendix B.2, Figure 4, and Figure 5) to illustrate how our model works.

There is a common question about the zero-shot evaluation, which we address here. We address other individual questions in the rely to each reviewer separately.

## Zero-shot evaluation:

1. We want to clarify that our training set does not include any example QAs (see Section 3.4). As explained in Section 3.4, we provide a training set of people’s typical behaviors, so that a model can try to learn how a person updates the mental state and actions to pursue a goal. Following the standard evaluation protocol in recent studies on ToM QA evaluation (e.g., Shapira et al., 2023), we evaluate a model on our test set in a zero-shot manner; that is, the model never sees any QA examples. This ensures that a model really learns the causal relationships between a person’s mental state and the actions, rather than learning patterns in the training QAs.
To evaluate how a baseline model trained on the training data can perform in our benchmark, we have finetuned VideoLLaMA 2 on our training data following the video instruction finetuning method used in the original VideoLLaMA 2 paper (see more details in Section 5.3 and Table 6 Appendix B.1). We use this finetuned Video-LLaMA 2 (13B) to evaluate two baselines: Video-LLaMA 2 (FT) and GPT-4 + captions (FT). Note GPT-4 + captions (FT) uses the captions generated by Video-LLaMA 2. We found that after finetuning, Video-LLaMA 2 performs moderately better in a few simpler question types, most notably true belief (Type 1.1), but its overall performance is still not better than chance.

2. Moreover, we found that few-shot prompting does not improve baselines meaningfully and consistently across all types and conditions ( they still perform no better than chance). The only exception is GPT-4 in the text-only condition, which performs better in simpler types (e.g., Type 1.3) with few-shot prompting but not in harder types (e.g., Type 1.2).

3. Finally, in our ablated study (Table 3 in Appendix B.1), we found that our model using pretrained GPT-J or LLaMA without finetuning also performed significantly better than using the original GPT-J and LLaMA alone; it even outperformed GPT-4 which is a much larger model. This shows the advantage of our method even without any training.

In sum, (1) our evaluation follows standard protocol in ToM evaluation; (2) finetuning pretrained models using common methods does not work as well as our method; (3) few-shot and chain-of-thought prompting does not offer significant improvement across different question types and modalities for baselines; and (4) our method without finetuning LMs can already outperform the baselines.

---

### Author Response · Authors · 2023-11-21
**Looking forward to your feedback**

Dear reviewers:

Thank you again for all the comments and suggestions. We have substantially revised the paper and enhanced the evaluation with more baselines and experiments. We have also responded to individual comments and questions. The discussion phase is going to end soon. We would love to hear your feedback and address any additional concerns before the discussion ends.

Best,

Authors

---

### Meta-Review · Area_Chair_uZnz · 2023-12-14

**Metareview:**

This paper introduces the BIP-ALM (Bayesian Inverse Planning Accelerated by Language Models) method for evaluating Theory of Mind (ToM) in multimodal contexts. The authors have done significant efforts to address the reviewers' concerns and expanded their experimental scope, incorporating evaluations with different model sizes, few-shot and chain-of-thought prompting, and finetuning on additional datasets.

However, despite these improvements, key concerns still linger. The reviewers' points about the necessity for clearer differentiation from existing VQA benchmarks, and a more robust justification for the chosen methodology and dataset design, remain partially addressed. Moreover, the comparison between the proposed BIP-ALM method and existing baselines requires further refinement to ensure fairness and comprehensiveness. While the authors have made notable progress, especially in clarifying the zero-shot evaluation protocol and demonstrating the robustness of their approach, the paper as it currently stands does not fully meet the acceptance criteria. The potential of the paper is evident, yet further development is needed to resolve the outstanding issues. The authors are encouraged to improve the paper according to the reviews for future resubmission.

**Justification For Why Not Higher Score:**

Save as above.

**Justification For Why Not Lower Score:**

N/A

---

### Decision · Program_Chairs · 2024-01-16

Reject